# Complexity of Derivative-Free Policy Optimization for Structured $\mathcal{H}_\infty$ Control

**Xingang Guo**
ECE, CSL
UIUC
xingang2@illinois.edu

**Darioush Keivan**
MSE, CSL
UIUC
dk12@illinois.edu

**Geir Dullerud**
MSE, CSL
UIUC
dullerud@illinois.edu

**Peter Seiler**
EECS
UMich
pseiler@umich.edu

**Bin Hu**
ECE, CSL
UIUC
binhu7@illinois.edu

## Abstract

The applications of direct policy search in reinforcement learning and continuous control have received increasing attention. In this work, we present novel theoretical results on the complexity of derivative-free policy optimization on an important class of robust control tasks, namely the structured $\mathcal{H}_\infty$ synthesis with static output feedback. Optimal $\mathcal{H}_\infty$ synthesis under structural constraints leads to a constrained nonconvex nonsmooth problem and is typically addressed using subgradient-based policy search techniques that are built upon the concept of Goldstein subdifferential or other notions of enlarged subdifferential. In this paper, we study the complexity of finding $(\delta, \epsilon)$-stationary points for such nonsmooth robust control design tasks using policy optimization methods which can only access the zeroth-order oracle (i.e. the $\mathcal{H}_\infty$ norm of the closed-loop system). First, we study the exact oracle setting and identify the coerciveness of the cost function to prove high-probability feasibility/complexity bounds for derivative-free policy optimization on this problem. Next, we derive a sample complexity result for the multi-input multi-output (MIMO) $\mathcal{H}_\infty$-norm estimation. We combine this with our analysis to obtain the first sample complexity of model-free, trajectory-based, zeroth-order policy optimization on finding $(\delta, \epsilon)$-stationary points for structured $\mathcal{H}_\infty$ control. Numerical results are also provided to demonstrate our theory.

## 1 Introduction

Policy optimization techniques have received increasing attention due to their impressive performance in reinforcement Learning (RL) and continuous control tasks [65, 48, 64, 45]. Despite the empirical successes, the theoretical properties of policy-based RL methods have not been fully understood, even on relatively simple linear control benchmarks. This has motivated a line of recent work developing sample complexity theory of model-free policy optimization on benchmark linear control problems such as linear quadratic regulator (LQR) [22, 47, 72, 50, 24, 33], stabilization [57, 56], linear robust/risk-sensitive control [30, 76, 77], Markov jump linear quadratic control [59], and distributed LQR [44]. These existing sample complexity results heavily rely on the fact that the cost functions in these benchmark problems are differentiable over the entire feasible set, and hence cannot be applied to cover the $\mathcal{H}_\infty$ robust control setting where the objective function is nonsmooth in the first place [2, 3, 18]. In this paper, we make a meaningful initial step to bridge this gap by

37th Conference on Neural Information Processing Systems (NeurIPS 2023).

developing a novel complexity theory for an important class of $\mathcal{H}_\infty$ robust control problems, namely the structured $\mathcal{H}_\infty$ synthesis with static output feedback.

$\mathcal{H}_\infty$ synthesis is arguably the most fundamental paradigm for robust control [80, 20]. Therefore, it is important to understand the complexity of policy optimization on such problems. In this work, we will develop complexity theory for derivative-free policy optimization on the structured static output-feedback $\mathcal{H}_\infty$ synthesis problem [3]. Notice that this is an important class of $\mathcal{H}_\infty$ control problems with practical importance [2, 3, 4, 31, 12, 18] due to the following reasons.

- Structured control refers to fixing the structure of the controller/policy without using the order information of the plant to be controlled. In practice, the state dimension of the true system is typically unknown (e.g. the system is assumed to be rigid body dynamics, but there will always be flexible modes with unknown state order for the true plant due to elasticity or other unmodeled dynamics). It can be very challenging to learn the exact order of the plant [55, 61]. Therefore, it is preferred to fix the structure of the controller/policy beforehand.

- Static output feedback is one of the most important forms of structured controller used in control, and it has a long history dating back to [42, 67, 49, 60]. Static output feedback has been extensively studied and applied in various areas, including aerospace [23], robotics [25], and chemical engineering [1]. In addition, the static output-feedback setting covers several important problems as special cases such as distributed controller design [14, 34] and proportional–integral–derivative (PID) control [73].

Studying the complexity of policy optimization methods on structured $\mathcal{H}_\infty$ synthesis is of great importance for several reasons. First, this problem cannot be solved using convexification techniques [62, 7, 26, 63] and hence it is natural to pose this problem as policy optimization [2, 3]. This is significantly different from the full-order $\mathcal{H}_\infty$ control design which yields convex reparameterizations [80, 20]. Specifically, if the order (i.e. state dimension) of the plant is exactly known, one can use a dynamical controller whose order is the same as the order of the plant, and the resultant full-order $\mathcal{H}_\infty$ synthesis can be lifted as a convex optimization problem. However, if the plant order is unknown and the controller structure is fixed beforehand, the resultant structured $\mathcal{H}_\infty$ problem is non-convex by nature [2, 3]. Second, the objective function in structured $\mathcal{H}_\infty$ synthesis can be non-differentiable over important feasible points (e.g. stationary points), and hence the structured $\mathcal{H}_\infty$ synthesis is a nonconvex nonsmooth optimization problem by nature. In the model-based setting, various advanced nonconvex nonsmooth optimization methods have been developed to solve this problem [2, 3, 4, 31, 12, 18, 52, 13, 40], leading to practical toolboxes such as HIFOO [4, 31]. Some convergence analysis has also been recently developed in [32]. However, in the model-free setting, the sample complexity of this problem remains largely unknown. The nonsmoothness of the cost function raises new challenges in developing the complexity analysis. There is a gap between the existing sample complexity theory and the model-free policy optimization on nonsmooth $\mathcal{H}_\infty$ synthesis. Our work will bridge this gap via offering the first sample complexity result of model-free policy optimization on nonsmooth $\mathcal{H}_\infty$ robust control. Specifically, we study the complexity of derivative-free policy optimization for structured $\mathcal{H}_\infty$ static output feedback problem when we can only access the zeroth-order oracle. We consider both exact and inexact zeroth-order oracle settings. Our contributions are two-fold:

1. *Exact* zeroth-order oracle: We establish the coerciveness of the cost function and then provide high-probability feasibility and complexity bounds by leveraging the relationship between Goldstein's subdifferential and uniform smoothing. Specifically, we show that to find a $(\delta, \epsilon)$-stationary point, the required number of iterations is on the order of $\Theta(\frac{1}{\delta \epsilon^4})$.

2. *Inexact* zeroth-order oracle: We first derive sample complexity bounds for the MIMO $\mathcal{H}_\infty$ estimation, which are then combined to establish the high-probability sample complexity bounds on the model-free, trajectory-based zeroth-order policy optimization methods for structured $\mathcal{H}_\infty$ problem. In particular, we show that the sample complexity bounds for finding a $(\delta, \epsilon)$-stationary point under inexact oracle setting are on the order of $\Omega\left(\frac{1}{\delta^3 \epsilon^8}\right)$. To the best of our knowledge, this is the first sample complexity result for model-free policy optimization on nonsmooth $\mathcal{H}_\infty$ synthesis.

**More discussions on related work.** Our paper focuses on the sample complexity of model-free policy optimization. Other than sample complexity, many other results on optimization landscape and

algorithm convergence of policy optimization for control have also been recently developed in the literature [9, 21, 79, 78, 75, 74, 16, 58, 39, 37, 38, 69, 66, 36]. See [35] for a comprehensive survey.

## 2 Preliminaries and Problem Formulation

### 2.1 Notation

Let $\mathbb{R}^d$ denote the $d$-dimensional real vectors, $\mathbb{R}^{m \times n}$ denote the set of real matrices with dimension $m \times n$. For a matrix $A$, the notations $A^\mathsf{T}$, $\sigma_{\max}(A)(\|A\|)$, $\|A\|_F$, $\rho(A)$ denote its transpose, largest singular value (spectral norm), Frobenius norm, and spectral radius, respectively. The symbol $\mathbb{B}_r(x)$ denotes a closed Euclidean ball of radius $r$ around a point $x$. Consider a sequence $\mathbf{w} := \{w_0, w_1, \cdots\}$ with $w_t \in \mathbb{R}^{n_w}$ for all $t$. This sequence belongs to $\ell_2^{n_w}$ if $\sum_{t=0}^\infty \|w_t\|^2 < \infty$ where $\|w_t\|$ is the standard 2-norm of a vector. In addition, the 2-norm for $\mathbf{w} \in \ell_2^{n_w}$ is defined as $\|\mathbf{w}\|^2 := \sum_{t=0}^\infty \|w_t\|^2$. We say a function $f(x) = O(g(x))$ if $\limsup_{x\to\infty} \frac{f(x)}{g(x)} < \infty$, $f(x) = \Omega(g(x))$ if $\liminf_{x\to\infty} \frac{f(x)}{g(x)} > 0$, and $f(x) = \Theta(g(x))$ if $f(x) = O(g(x))$ and $f(x) = \Omega(g(x))$. Given a matrix sequence $\{P_k \in \mathbb{C}^{n \times n}\}_{k \in \mathbb{Z}_+}$, let $T(P)$ denote the infinite Toeplitz matrix $T(P) = (P_{i-j})_{i,j=0}^{\infty,i}$ and $T_N(P)$ denote the $N \times N$ block Toeplitz matrix $T_N(P) = (P_{i-j})_{i,j=0}^{N-1,i}$.

### 2.2 Static output feedback $\mathcal{H}_\infty$ control

Consider the following linear time-invariant (LTI) system

$$x_{t+1} = Ax_t + Bu_t + w_t, \ x_0 = 0 \tag{1}$$
$$y_t = Cx_t$$

where $x_t \in \mathbb{R}^{n_x}$ is the system state, $u_t \in \mathbb{R}^{n_u}$ is the control action, $w_t \in \mathbb{R}^{n_w}$ is the disturbance, and $y_t \in \mathbb{R}^{n_y}$ is the output measurement. We have $A \in \mathbb{R}^{n_x \times n_x}$, $B \in \mathbb{R}^{n_x \times n_u}$, $C \in \mathbb{R}^{n_y \times n_x}$, and $n_w = n_x$. The initial condition is fixed as $x_0 = 0$. We denote $\mathbf{w} := \{w_0, w_1, \cdots\}$[1].

In this work, we consider the static output feedback defined as $u_t = -Ky_t = -KCx_t$, where $K \in \mathbb{R}^{n_u \times n_y}$ is a constant matrix. Then the closed loop system is given by

$$x_{t+1} = (A - BKC)x_t + w_t, \ x_0 = 0. \tag{2}$$

The structured $\mathcal{H}_\infty$ synthesis with static output feedback is defined as the following minimax problem

$$\min_{K \in \mathcal{K}} \max_{\mathbf{w}:\|\mathbf{w}\|\le 1} \sum_{t=0}^\infty x_t^\mathsf{T}(Q + C^\mathsf{T}K^\mathsf{T}RKC)x_t, \tag{3}$$

where $\mathcal{K}$ is the set of all linear static output feedback stabilizing policies, i.e. $\mathcal{K} = \{K \in \mathbb{R}^{n_u \times n_y} : \rho(A - BKC) < 1\}$ and we consider the standard quadratic cost function with worst case disturbance $\mathbf{w}$ that satisfies the $\ell_2$ norm bound $\|\mathbf{w}\| \le 1$. This is different than the conventional Linear Quadratic Regulator (LQR) problem where only the stochastic disturbance is considered. Throughout this paper, we also adopt the following standard assumption [21].

**Assumption 2.1.** There exists a $K^0 \in \mathcal{K}$. The matrices $Q$ and $R$ are positive definite. The matrix $C$ is full row rank.

In general, the problem of finding a stablizing static output feedback policy is known to be NP-hard [5]. For the developments of policy optimization theory, it is standard to assume that such an initial stabilizing policy is available.

In the frequency domain, we can show that the above cost function (3) is equivalent to the $\mathcal{H}_\infty$ norm of the associated closed-loop transfer function [3]. Specifically, the structured $\mathcal{H}_\infty$ static output-feedback control problem can be formulated as

$$\min_{K \in \mathcal{K}} J(K), \tag{4}$$

---

[1]Our results in this work can be generalized to the case with measurement noise, i.e., $y_t = Cx_t + v_t$. See more discussions on this extension in the supplementary material.

where $J(K)$ is the $\mathcal{H}_\infty$ norm of the associated closed-loop transfer function that can be calculated as

$$J(K) = \sup_{\omega \in [0, 2\pi]} \sigma_{\max}\left((Q + C^\mathsf{T} K^\mathsf{T} R K C)^{\frac{1}{2}}(e^{j\omega} I - A + BKC)^{-1}\right). \tag{5}$$

More details on the derivation of (5) can be found in the supplementary material. We emphasize that the policy search problem (4) is a nonconvex nonsmooth optimization problem. There are two sources of nonsmoothness for the cost (5). Specifically, computing the largest singular value and taking supremum over the frequency $\omega \in [0, 2\pi]$ can both be nonsmooth. Consequently, the objective function (5) can be non-differentiable over some important feasible points [2, 3, 4, 31, 12, 18].

### 2.3 Subgradient methods in the model-based setting

A key concept in the nonconvex nonsmooth optimization theory is the so-called Clarke subdifferential [15, 43]. A function $J : \mathcal{K} \to \mathbb{R}$ is said to be locally Lipschitz if for any bounded set $S \subset \mathcal{K}$, there exists a constant $L$ such that $|J(K) - J(K')| \le L\|K - K'\|_F$ for all $K, K' \in S$. It is well known that the closed-loop $\mathcal{H}_\infty$ cost (5) is locally Lipschitz over the set of stabilizing controllers [3]. For a locally Lipschitz function, the Clarke subdifferential exists and is defined as $\partial J(K) := \text{conv}\{\lim_{l \to \infty} \nabla J(K^l) : K^l \to K, K^l \in \text{dom}(\nabla J) \subset \mathcal{K}\}$, where conv stands for the convex hull [15]. In the model-based setting, structural $\mathcal{H}_\infty$ synthesis is typically solved using advanced nonsmooth optimization algorithms that generate good descent directions via enlarging the Clarke subdifferential. Two main types of algorithms are briefly reviewed as follows.

**Goldstein's subgradient method and gradient sampling.** The main workhorse for the HIFOO toolbox is the gradient sampling method [11], which is developed based on the concept of Goldstein subdifferential [28]. Specifically, the Goldstein $\delta$-subdifferential for a point $K \in \mathcal{K}$ is defined as

$$\partial_\delta J(K) := \text{conv}\{\cup_{K' \in \mathbb{B}_\delta(K)} \partial J(K')\}, \tag{6}$$

which implicitly requires $\mathbb{B}_\delta(K) \subseteq \mathcal{K}$. It is well known that the minimum norm element of the Goldstein subdifferential generates a good descent direction, i.e. $J(K - \delta H/\|H\|_F) \le J(K) - \delta\|H\|_F$ for $H$ being the minimum norm element of $\partial_\delta J(K)$ [28]. Although calculating the exact minimum norm element from the Goldstein subdifferential can be difficult, one can still estimate a good descent direction from approximating $\partial_\delta J(K)$ as the convex hull of randomly sampled gradients over $\mathbb{B}_\delta(K)$. This leads to the gradient sampling method.

**Frequency-domain methods.** Another popular technique is generateing the descent directions via enlarging the Clarke subdifferential in the frequency domain [17, 54]. Such an enlargement method relies on standard chain rules to exploit the fact that the closed-loop $\mathcal{H}_\infty$ cost can be rewritten as a composition of a smooth mapping and a convex norm. The `Matlab` function `Hinfstruct` from the robust control package [3] is developed based on such a frequency-domain technique.

### 2.4 Randomized smoothing

Our analysis relies on the concept of randomized smoothing techniques, which have been widely used in convex/nonconvex optimization problems [19, 27]. The smoothed version of $J(K)$ via uniform randomized smoothing is defined as below.

**Definition 2.2.** Consider an $L$-Lipschitz function $J$ (possibly nonconvex and nonsmooth) and a uniform distribution $\mathbb{P}$ on $\{U \in \mathbb{R}^{n_u \times n_y} : \|U\|_F = 1\}$. Then the smoothed version of $J$ is defined as $J_\delta$ is defined as

$$J_\delta(K) = \mathbb{E}_{U \sim \mathbb{P}}[J(K + \delta U)] \tag{7}$$

The above definition requires both $K$ and $K + \delta U$ to belong to the feasible set $\mathcal{K}$ for all $U \sim \mathbb{P}$. Very recently, [46] established the relationship between Goldstein subdifferential and uniform smoothing. We briefly restate such connections below.

**Lemma 2.3.** *Suppose $J(K)$ is L-Lipschitz over some bounded set $\mathcal{S} \subset \mathcal{K}$, and for any $K \in \mathcal{S}$, we have $K + \delta U \in \mathcal{K}$ for all $U$ with $\|U\|_F = 1$ such that $J_\delta(K)$ is well defined (see Definition 2.2). Let $\partial_\delta J(K)$ be the Goldstein $\delta$-subdifferential at $K$, then for any $K \in \mathcal{S}$, we have:*
*(i) $|J(K) - J_\delta(K)| \le \delta L$,*
*(ii) $J_\delta$ is differentiable and L-Lipschitz with the $\frac{cL\sqrt{d}}{\delta}$-Lipschtiz gradient, where $d = n_y n_u$ is the problem dimension and $c > 0$ is a constant,*
*(iii) $\nabla J_\delta(K) \in \partial_\delta J(K)$.*

This lemma is essentially [46, Proposition 2.3, Theorem 3.1]. Based on the definition of Goldstein $\delta$-subdifferential (6), we say a point $K$ is a $(\delta, \epsilon)$-stationary point if $\text{dist}(0, \partial_\delta J(K)) \leq \epsilon$. Then Lemma 2.3 (iii) implies that if $K$ is a $\epsilon$-stationary point of $J_\delta(K)$ (i.e., $\|\nabla J_\delta(K)\|_F \leq \epsilon$), then $K$ is also a $(\delta, \epsilon)$-stationary point of the original function $J(K)$.

Suppose that we can compute $\nabla J_\delta(K)$, then we can simply apply gradient descent based on the gradient of $J_\delta(K)$

$$K^{t+1} = K^t - \eta \nabla J_\delta(K^t) \tag{8}$$

to find a $\epsilon$-stationary point of $J_\delta(K)$. However, how to choose the stepsize $\eta$ and smooth radius $\delta$ to guarantee the feasibility and convergence to an $(\delta, \epsilon)$-stationary points of the original cost function (5) is unknown. We will discuss this setup more in Section 3.2.

### 2.5 Problem formulation: two zeroth-order oracle assumptions

In this paper, we aim to minimize the cost function $J(K)$ in the policy space directly via derivative-free methods. Specifically, we consider two different zeroth-order oracle settings:

1. *Exact* zeroth-order oracle: This oracle assumption is standard for zeroth-order optimization literature and natural for the model-based control setting. In particular, we assume that we can exactly calculate $J(K)$ (which is the closed-loop $\mathcal{H}_\infty$ norm) for every stabilizing $K$. When the system dynamics are known, such an oracle is available since the closed-loop $\mathcal{H}_\infty$ norm can be efficiently calculated using existing robust control packages in MATLAB (currently, the state-of-the-art techniques for model-based $\mathcal{H}_\infty$ norm calculations rely on using the relation between the singular values of the transfer function matrix and the eigenvalues of a related Hamiltonian matrix [6, 8]). Under the same oracle setting, the non-derivative sampling method has been successfully applied to state-feedback $\mathcal{H}_\infty$ control problem in [32] without complexity guarantee. In this work, we close this gap by showing that, for constrained policy optimization problem (4), our algorithm returns a $(\delta, \epsilon)$-stationary point with high-probability feasibility/complexity bounds.

2. *Inexact* zeroth-order oracle: This oracle assumption is relevant for the model-free learning-based control setting, where the system dynamics are unknown, and $J(K)$ (the closed-loop $\mathcal{H}_\infty$ norm) can only be estimated from the input/output data of a black-box simulator of the underlying system. In particular, we use the model-free time-reversal power-iteration-based $\mathcal{H}_\infty$ estimation from [70] to serve as the inexact oracle for $J(K)$. Despite the existence of such algorithms, the prior literature lacks sample complexity bounds for general MIMO systems. Therefore, we presented the first sample complexity result for $\mathcal{H}_\infty$ norm estimation for general MIMO systems. Building upon this, we obtain the first sample complexity results for model-free policy optimization of $\mathcal{H}_\infty$ control with noisy function values.

## 3 Main Results

In this section, we present our main result for the exact zeroth-order oracle case. We will start with the optimization landscape of the policy optimization problem (4).

### 3.1 Optimization landscape

**Proposition 3.1.** *The set $\mathcal{K} = \{K : \rho(A - BKC) < 1\}$ is open and nonconvex. It can be disconnected. In general, it can be either bounded or unbounded. The cost function (5) is locally Lipschitz over $\mathcal{K}$ and hence is continuous in $K$.*

The proof of the above proposition can be found in [10, 22, 3]. Next, we identify the coerciveness of the cost function $J(K)$.

**Lemma 3.2.** *The $\mathcal{H}_\infty$ objective function $J(K)$ defined by (5) is coercive over the set $\mathcal{K}$ in the sense that for any sequence $\{K^l\}_{l=1}^\infty \subset \mathcal{K}$ we have*

$$J(K^l) \to +\infty$$

*if either $\|K^l\|_F \to +\infty$, or $K^l$ converges to an element in the boundary $\partial \mathcal{K}$.*

It is worth mentioning that the above lemma relies on the assumption that both $Q$ and $R$ are positive definite. The formal proof of Lemma 3.2 is deferred to the supplementary material. The continuity and coerciveness of $J(K)$ directly lead to the following results.

**Lemma 3.3.** *Consider the $\mathcal{H}_\infty$ output feedback policy search problem* (4) *with the objective function $J(K)$ defined in* (5). *Under Assumption 2.1, for any $\gamma > J^*$, the sublevel set defined as $\mathcal{S}_\gamma := \{K \in \mathcal{K} : J(K) \le \gamma\}$ is compact.*

*Remark* 3.4. Based on Lemma 3.3 and the fact that the set $\mathcal{K}$ is open, we can show that there is a strict separation between the sublevel set $\mathcal{S}_\gamma$ and $\partial\mathcal{K}$. It is obvious that $\partial\mathcal{K}$ is closed. Since $\mathcal{S}_\gamma$ is compact and $\mathcal{S}_\gamma \cap \partial\mathcal{K} = \varnothing$, we have $\mathrm{dist}(\mathcal{S}_\gamma, \partial\mathcal{K}) > 0$.

## 3.2 Warm-up: convergence of smoothed function

In this section, we assume that the exact gradient oracle $\nabla J_\delta(K)$ is available for each $K \in \mathcal{K}$ such that $J_\delta(K)$ is well defined. Then we analyze the finite-time convergence and feasibility of the vanilla gradient descent method (8). To this end, let $K^0 \in \mathcal{K}$ be an arbitrary feasible initial controller. Without loss of generality, define two feasible sets:

$$\mathcal{S}^0 := \{K|J(K) \le 50J(K^0)\}, \ \mathcal{S}^1 := \{K|J(K) \le 100J(K^0)\}. \tag{9}$$

Denote $\Delta_0 = \mathrm{dist}(\mathcal{S}^0, \partial\mathcal{K})$ and $\Delta_1 = \mathrm{dist}(\mathcal{S}^1, \partial\mathcal{K})$. Let $L_0$ and $L_1$ be the Lipschitz constants of $J(K)$ associated with the sublevel set $\mathcal{S}^0$ and $\mathcal{S}^1$, respectively. It is obvious that $\Delta_0 \ge \Delta_1$ and $L_0 \le L_1$. In addition, we have $\Delta = \mathrm{dist}(\mathcal{S}^0, \partial\mathcal{S}^1) > 0$ by Remark 3.4.

*Remark* 3.5. Since $J(K)$ is a continuous function of $K$ by Proposition 3.1 and the sublevel sets $\mathcal{S}^0$ and $\mathcal{S}^1$ are compact by Lemma 3.3, this implies that there exists a constant $\xi > 0$ such that for any $K \in \mathcal{S}^0$, and $K'$ with $\|K - K'\|_F \le \xi$, we have $K' \in \mathcal{S}^1$.

Now we are ready to state the following result.

**Theorem 3.6.** *Let $K^0 \in \mathcal{K}$ be an arbitrary feasible initial controller and suppose that $J_\delta(K)$ defined as in* (7) *is $L_1$-Lipschitz with the $\frac{cL_1\sqrt{d}}{\delta}$-Lipschtiz gradient on the sublevel set $\mathcal{S}^1$. Choose $\delta = \min\{\Delta_1, \frac{49J(K^0)}{2L_1}\}$ and $\eta = \min\{\frac{\xi}{L_1}, \frac{\delta}{cL_1\sqrt{d}}\}$. Then the iterative method* (8) *stays in $\mathcal{K}$ and we have:*

$$\min_{t=0,1,\cdots,T-1} \|\nabla J_\delta(K^t)\|_F^2 \le \frac{2(J_\delta(K^0) - J^*)}{\eta T}. \tag{10}$$

*In other words, we have $\min_{0 \le t \le T-1} \|\nabla J_\delta(K^t)\|_F \le \epsilon$ after $T = O(\frac{1}{\epsilon^2})$.*

The proof of Theorem 3.6 can be found in the supplementary material. Since we have a constrained optimization problem, it is crucial to guarantee the feasibility of the gradient descent method (8). The careful choice of smooth radius $\delta$ and step size $\eta$ ensures that $K^t$ stays inside the feasible set.

## 3.3 Analysis for the exact oracle case

In this section, we consider the exact zeroth-order oracle setup, where we can obtain the exact $\mathcal{H}_\infty$ norm of the closed-loop system for a given $K \in \mathcal{K}$. One straightforward extension of (8) is to compute an unbiased estimation of the gradient $\nabla J_\delta(K^t)$ via the zeroth-order oracle. Then we can perform the one-step gradient descent. In particular, we analyze the feasibility and convergence of the Algorithm 1 in the following result.

**Theorem 3.7.** *Let $K^0 \in \mathcal{K}$ be an arbitrary feasible initial controller and suppose $J(K)$ is $L_1$-Lipschtiz with the $\frac{cL_1\sqrt{d}}{\delta}$-Lipschtiz gradient on the sublevel set $\mathcal{S}^1$. Let $1 \le \upsilon \le 80$ and suppose we choose*

$$\delta = \min\{\Delta_1, \Delta, \frac{J(K^0)}{L_1}\}, \ T = \frac{\upsilon\delta J(K^0)}{50\eta^2 cd^{3/2}L_1^3}, \ \eta = \min\{\frac{2\delta\xi}{d(100J(K^0) - J^*)}, \frac{\delta\epsilon^2}{3500cd^{3/2}L_1^3}\}.$$

*Then the following two statements hold:*
*(1) the controllers $\{K^t\}_{t=0}^{T-1}$ generated by Algorithm 1 are all stabilizing with probability at least $0.95 - 0.01\upsilon$.*
*(2) the output of Algorithm 1 $K^R$ satisfies*

$$\min\{\|H\|_F : H \in \partial_\delta J(K^R)\} \le \epsilon \tag{11}$$

*with probability at least $0.87 - 0.17\upsilon^{-\frac{1}{2}} - 0.01\upsilon$.*

---

**Algorithm 1:** Derivative-Free Methods for Policy Optimization Problem (4)

---

**Require:** feasible initial point $K^0 \in \mathbb{R}^{n_u \times n_y}$, stepsize $\eta > 0$, problem dimension $d := n_u \times n_y \geq 1$, smoothing parameter $\delta$ and iteration number $T \geq 1$.

**for** $t = 0, 1, \cdots, T - 1$ **do**

    Sample $W^t \in \mathbb{R}^{n_u \times n_y}$ uniformly at random over matrices such that $\|W\|_F = 1$.

    Compute $g^t = \frac{d}{2\delta}(J(K^t + \delta W^t) - J(K^t - \delta W^t))W^t$.

    Update $K^{t+1} = K^t - \eta g^t$.

**end for**

**Output:** $K^R$ where $R \in \{0, 1, 2, \cdots, T - 1\}$ is uniformly sampled.

---

The proof of Theorem 3.7 can be found in the supplementary material. Now we provide some discussions regarding Theorem 3.7 in order:

**Probability Bounds:** Statement 1 suggests that as $T$ increases, the probability that all the generated controllers are stabilizing will decrease. This is because our algorithm uses a zeroth-order oracle to build an estimator of the smoothed function gradient. As $T$ increases, the biases and variance of the gradient estimation accumulate, resulting in a larger failure probability. In addition, Statement 2 suggests that as $T$ increases, the probability of finding a $(\delta, \epsilon)$-stationary point will first increase and then decrease. Indeed, when $T$ is too small, more iterations will improve the performance of the generated controllers, while for large $T$, the probability of generating unstable controllers becomes dominant. In addition, the constant factors in the probability bounds are not restrictive, they can be further improved by e.g., increasing the level of $\mathcal{S}^0$, using smaller step size $\eta$ or using smaller smooth radius $\delta$ in the analysis.

**Feasibility:** Statement 1 states that all the output controllers $\{K^t\}_{t=0}^{T-1}$ are feasible with high probability under the choice of algorithm parameters. It is worth mentioning that, to ensure the feasibility of the iterates $\{K^t\}_{t=0}^{T-1}$, we need to show that $\{J_\delta(K^t)\}_{t=0}^{T-1}$ are well defined so that we can apply the smoothness property of $J_\delta(K)$ in our theoretical analysis. Furthermore, this also guarantees that $\{J(K^t \pm \delta W^t)\}_{t=0}^{T-1}$ used in Algorithm 1 are well defined. This turns out to be a nontrivial task and one needs to design algorithm parameters carefully to resolve the feasibility issue. More discussions will be provided in the end of this section.

**Output Controller:** Since our cost function $J(K)$ is nonconvex, we consider an output controller $K^R$ uniformly sampled from $\{K^t\}_{t=0}^{T-1}$. Such random output technique has been used in smooth/nonsmooth nonconvex optimization problems for theoretical analysis [27, 46, 44]. In particular, Statement 2 implies that the output controller $K^R$ is a $(\delta, \epsilon)$-stationary point with high probability. In practical implementation, one can just select the iterate that gives the lowest cost $J(K^t)$ for $t = 1, \cdots, T$. Our numerical experiments suggest that selecting $K^T$ often yields satisfactory performance (see Section 5).

**Sample Complexity:** For sufficiently small $\epsilon$, the number of iterations to guarantee (11) is given by:

$$T = \Theta\left(\frac{d^{\frac{3}{2}}L_1^3}{\delta \epsilon^4}\right), \tag{12}$$

where $d \geq 1$ is the problem dimension, $L_1$ is the Lipschtiz parameter. We also ignore the numerical constants since they are conservative and not restrictive.

Finally, we close this section by highlighting the main technical contributions of Theorem 3.7 and 3.6. In our control setup, unlike the unconstrained optimization problems [46], we need to ensure that the iterate $K^t$ and the perturbed iterate $K^t \pm \delta W^t$ stay within a non-convex feasible set $\mathcal{K}$. Previous work on policy optimization theory of $\mathcal{H}_\infty$ control addresses this feasibility issue via using the coerciveness of $J(K)$ and mainly relies on the fact that $J(K)$ is a barrier function on the non-convex set of stabilizing policies [32]. In particular, such previous results rely on model-based algorithms (such as Goldstein's subgradient method) which can decrease the value of $J(K)$ directly. However, the zeroth-order policy search can only decrease the value of the smoothed function $J_\delta(K)$, which is not coercive over the non-convex feasible set and hence cannot be used as a barrier function. Importantly, the descent of $J_\delta(K)$ does not imply the descent of the original function value and hence cannot ensure feasibility by itself. In Theorem 3.7 and 3.6, by carefully choosing the smooth radius

$\delta$ and step size $\eta$, we manage to show that the iterate $K^t$ and the perturbed iterate $K^t \pm \delta W^t$ stay within a non-convex feasible set with high probability.

# 4 Sample Complexity for the Model-Free Case

In this section, we assume that the system dynamics in (1) are unknown and we can only use samples to obtain an estimation of the cost function $\hat{J}(K)$ with some error $\hat{J}(K) = J(K) + \zeta(K)$. To obtain an end-to-end sample complexity result of Algorithm 1, we first derive the sample complexity of $\mathcal{H}_\infty$ norm estimation for the general MIMO systems such that $\zeta(K) \leq \kappa$ for all $K \in \mathcal{K}$.

## 4.1 Complexity of $\mathcal{H}_\infty$ norm estimation for MIMO systems

The $\mathcal{H}_\infty$ norm estimation method via input/output data can be roughly categorized into two major approaches: (a) try to find the worst case $\ell_2$-norm signal using power-iteration algorithm [71, 53]. (b) discretizing the interval $[0, 2\pi]$ and search for the maximizing frequency using multi-armed bandit [51]. However, there are no sample complexity results of the aforementioned methods for the general MIMO system. In this section, we analyze the $\mathcal{H}_\infty$ norm estimation method proposed in [53] and establish its sample complexity bounds for the general MIMO systems.

To this end, let $P(z) = (Q + C^T K^T RKC)^{1/2}(zI - A + BKC)^{-1}$ be the transfer function associated with the system (1) to the cost function (3), let $T(P)$ denote the corresponding Toeplitz (convolution) operator and $T_N(P)$ be the $N \times N$ upper-left submatrix of $T(P)$. Then it is known that $J(K) = \|T(P)\|$ and $\|T_N(P)\| \to \|T(P)\|$ as $N \to \infty$. With large enough $N$, the largest singular value of the $T_N(P)$ can be used as a reasonable estimation of $J(K)$, which can be approximated by running the power iteration algorithm $n$ steps [2]. Therefore, there are two error terms based on this approach. The first term $\zeta_1(K)$ comes from the difference between $\|T_N(P)\|$ and $\|T(P)\|$, and the second term $\zeta_2(K)$ is induced by computing the largest singular value of $T_N(P)$ via the power iteration algorithm:

$$|\zeta(K)| = |J(K) - \hat{J}(K)| \leq |J(K) - \|T_N(P)\|| + |\|T_N(P)\| - \hat{J}(K)| = \zeta_1(K) + \zeta_2(K), \quad (13)$$

where $\hat{J}(K)$ is the approximated largest singular value of $T_N(P)$ and hence an estimation of $J(K)$. Therefore, if we can choose $N$ and $n$ large enough such that $\zeta_1(K) \leq \kappa/2$ and $\zeta_2(K) \leq \kappa/2$, we will have $\zeta(K) \leq \kappa$ holds by (13). The finite-time condition on $N$ has been established in [68] for single input single output (SISO) systems only. In the following result, we extend the result in [68] to the general MIMO system. Furthermore, we provide the finite-time condition on the number of power iteration iterations $n$.

**Theorem 4.1.** *Let* $P(z) = \sum_{k=0}^\infty P_k z^{-k}$ *be the corresponding transfer function of system* (1) *to the cost function* (3) *with stability radius* $\iota \in (0, 1)$. *And choose* $\gamma \in (\iota, 1)$ *and* $D \geq \sigma_{max}(P_0)$. *Let* $\|P^\gamma\|_\infty$ *denotes the* $\mathcal{H}_\infty$ *norm of the system* $P^\gamma(z) := \gamma P(\gamma z)$. *Then for* $N \geq 3$, *we have*

$$|\hat{J}(K) - J(K)| \leq \zeta_1(K) + \zeta_2(K) \quad (14)$$

*where*

$$\zeta_1(K) = C_1 \frac{D\|P^\gamma\|_\infty(1-\gamma^2) + \|P^\gamma\|_\infty^2 \gamma}{\|P\|_\infty(1-\gamma)^4} \frac{1}{N^2} + C_2 \frac{\|P^\gamma\|_\infty^2}{\|P\|_\infty(1+\gamma)(1-\gamma)^5} \frac{1}{N^3}, \quad (15)$$

$$\zeta_2(K) = C_3 \|P\|_\infty n^{-\frac{2}{3}} \quad (16)$$

*are errors due to approximating* $T$ *by* $T_N$ *and using power iteration to estimate* $\|T_N\|$, *respectively. Here* $C_1 = 3\sqrt{2}\pi(2 + 3\pi^4), C_2 = 9\sqrt{2}\pi^2$ *are universal constants and* $C_3$ *depends on an angle between power iteration initialization and the eigenvector corresponding to the biggest eigenvalue of* $T_N^*(P)T_N(P)$.

The proof of Theorem 4.1 can be found in the supplementary material. Theorem 4.1 implies that the approximation error $|\zeta(K)| \leq \kappa$ is guaranteed by choosing

$$N \geq \inf_{\iota < \gamma < 1} \Omega\left(\frac{1}{(1-\gamma)^2}\sqrt{\frac{\|P^\gamma\|_\infty^2}{\|P\|_\infty}\frac{2}{\kappa}}\right), \quad n \geq \Omega\left(\frac{(2C_3)^{\frac{3}{2}}\|P\|_\infty^{\frac{3}{2}}}{\kappa^{\frac{3}{2}}}\right). \quad (17)$$

---

[2]For completeness, the pseudo-code of the power iteration algorithm is provided in the supplementary material.

## 4.2 Sample complexity of zeroth-order optimization

In this section, we consider that Algorithm 1 can only access the inexact zeroth-order oracle of the cost function $J(K)$. In this case, we do not have an unbiased estimation of $\nabla J_\delta(K^t)$. Nevertheless, we can make the estimation as small as possible and assume that the estimation error $\zeta(K)$ is uniformly upper bounded for all $K \in \mathcal{S}^1$:

$$|\zeta(K)| \leq \kappa, \ \forall K \in \mathcal{S}^1. \tag{18}$$

Clearly, the smaller $\kappa$ is, the larger power iteration number $n$ and approximation horizon $N$ are needed. We will specify the choice of $\kappa$ in the next result, where we obtain the feasibility/sample complexity bounds of Algorithm 1 via inexact zeroth-order oracle on finding a $(\delta, \epsilon)$-stationary point.

**Theorem 4.2.** *Let $K^0 \in \mathcal{K}$ be an arbitrary feasible initial controller and suppose $J$ is $L_1$-Lipschtiz on the sublevel set $\mathcal{S}^1$. Consider the Algorithm 1 with inexact zeroth-order oracle and suppose that the estimation error $\kappa \leq \frac{\delta \epsilon^2}{100 d L_1}$. Denote $\Gamma := c L_1 \sqrt{d}(16\sqrt{2\pi} d L_1^2 + (\frac{d\kappa}{\delta})^2)$. Let $1 \leq \upsilon \leq 25$ and suppose we choose*

$$\delta = \min\{\Delta_1, \Delta, \frac{J(K^0)}{L_1}\}, \ T = \frac{\upsilon \delta J(K^0)}{2\eta^2 \Gamma}, \ \eta = \min\{\frac{\delta \xi}{d\kappa(100 J(K^0) - J^*)}, \frac{\delta \epsilon^2}{100 \Gamma}\}.$$

*Then the following two statements hold:*
*(1) the controllers $\{K^t\}_{t=0}^{T-1}$ generated by Algorithm 1 with inexact zeroth-order oracle are all stabilizing with probability at least $0.95 - 0.03\upsilon$.*
*(2) the output of Algorithm 1 $K^R$ satisfies*

$$\min\{\|H\|_F : H \in \partial_\delta J(K^R)\} \leq \epsilon \tag{19}$$

*with probability at least $0.8 - 0.15\upsilon^{-\frac{1}{2}} - 0.03\upsilon$.*

The proof of Theorem 4.2 can be found in the supplementary material. For sufficiently small $\epsilon$, we can obtain the following lower bound on the number of iterations to guarantee (19):

$$T = \Omega\left(\frac{d^{\frac{3}{2}} L_1^3}{\delta \epsilon^4}\right), \tag{20}$$

Since we require $\kappa \leq \frac{\delta \epsilon^2}{100 d L_1}$, we have the following lower bounds on the power iteration number $n$ and approximation horizon $N$ by (17):

$$N = \Omega\left(\frac{d^{1/2} L_1^{1/2}}{\delta^{1/2} \epsilon}\right), \ \ n = \Omega\left(\frac{(d L_1)^{\frac{3}{2}}}{\delta^{\frac{3}{2}} \epsilon^3}\right).$$

Finally, by combining the above sample complexity bounds for MIMO $\mathcal{H}_\infty$ estimation with (20), the number of samples to guarantee (19) with high probability is given by:

$$nNT = \Omega\left(\frac{d^{7/2} L_1^5}{\delta^3 \epsilon^8}\right). \tag{21}$$

In this section, we consider the sample complexity of Algorithm 1 with the inexact oracle case. This is particularly relevant for the model-free control setting. Specifically, we are using imperfect estimates of $J(K)$ that are calculated using the model-free MIMO power iteration method. Therefore, an extra statistical error term appears in the iterations of zeroth-order policy optimization and requires special treatment. Such an extra term has not been considered in the literature on zeroth-order optimization for nonconvex nonsmooth problems. To address this extra technical difficulty, we first establish sample complexity bounds for $\mathcal{H}_\infty$ norm estimation of the general MIMO system (Theorem 4.1). Then we carefully propagate such sample complexity bounds to obtain an error bound for $\nabla J_\delta(K)$ in terms of $\epsilon$ and $\delta$. Built upon this, we demonstrate that Algorithm 1 remains effective even with an inexact oracle, ensuring the feasibility of the iterates while achieving finite-time sample complexity with high probability (Theorem 4.2).

# 5    Numerical Experiments

In this section, we present the numerical study to show the effectiveness of Algorithm 1. The left plot of Figure 1 displays the relative error trajectories of Algorithm 1 with an exact oracle for system dimensions $n_x = \{10, 50, 100\}$. The system is of form (1) with parameters $(A, B, C)$, where the entries of $(A, B, C)$ are sampled from a standard normal distribution $\mathcal{N}(0, 1)$. The results show that Algorithm 1 performs well even for larger-scale systems. In the middle plot, we compare the trajectories of Algorithm 1 with both exact and inexact zeroth-order oracle. It can be observed that the inexact oracle case closely tracks the performance of the exact oracle case. The right plot illustrates the zeroth-order complexity of the exact oracle case with varying $\epsilon$. As shown, the number of oracle calls increases as $\epsilon$ decreases. It is important to note that the complexity bounds derived in (12) are not tight. We conjecture that by leveraging other advanced properties of the cost function (5), further improvements in complexity can be achieved.

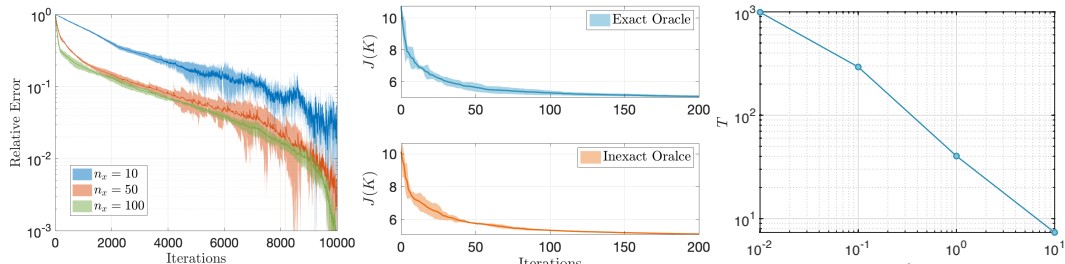

Figure 1: Left: The relative error trajectories of Algorithm 1 with exact zeroth-order oracle and $n_x = \{10, 50, 100\}$, the solid lines represent the mean values and the shade represents 98% confidence intervals; Middle: The trajectory of Algorithm 1 with exact and inexact zeroth-order oracle; Right: Number of zeroth-order oracle calls required to find a $(\delta, \epsilon)$-stationary point with varying $\epsilon$.

Table 1: Comparison of Algorithm 1 with model-based methods

| Example | $(n_x, n_u, n_y)$ | HIFOO | Hinfstruct | Algorithm 1 **(Ours)** |
|---------|-------------------|-------|------------|------------------------|
| AC15    | $(4, 2, 3)$       | 15.2919             | 15.2              | 15.4141               |
| HF2D11  | $(5, 2, 3)$       | $7.7237 \times 10^4$ | $7.72 \times 10^4$ | $7.7223 \times 10^4$  |
| DLR2    | $(40, 2, 2)$      | $4.0066 \times 10^3$ | $4.01 \times 10^3$ | $4.0094 \times 10^3$  |
| HE4     | $(8, 4, 6)$       | 22.8382             | 22.8              | 22.8538               |

In addition, we conducted a comparison of our derivative-free method with the model-based methods HIFOO and Hinfstruct, using several benchmark examples from COMPl$_e$ib [41]. Table 1 shows the corresponding optimal closed-loop $\mathcal{H}_\infty$ norm. It can be seen that our derivative-free method yields comparable results with the model-based packages even without the knowledge of system models. More details about the numerical experiments can be found in the supplementary material.

# 6    Conclusion and Future Work

This paper presents the feasibility and complexity bounds for the derivative-free policy optimization method on the structured $\mathcal{H}_\infty$ synthesis, considering both exact and inexact zeroth-order oracles. Despite the fact that this structured $\mathcal{H}_\infty$ synthesis is a constrained nonconvex nonsmooth optimization problem, we leverage the intriguing connections between randomized smoothing and $(\delta, \epsilon)$-stationarity, enabling the first sample analysis for model-free, trajectory-based, zeroth-order policy optimization in structured $\mathcal{H}_\infty$ synthesis. One limitation of this work is the uncertain tightness of the sample complexity bounds obtained.

For future studies, it is important to tighten the sample complexity results by exploring more properties of $\mathcal{H}_\infty$ control problems. In addition, it is interesting to explore the behavior of the structured $\mathcal{H}_\infty$ problem with a dynamic output feedback controller.

## Acknowledgement

The work of Xingang Guo and Bin Hu is generously supported by the NSF award CAREER-2048168 and the IBM/IIDAI award 110662-01. The work of Darioush Kevian and Geir Dullerud was supported by NSF under Grant ECCS 1932735. The work of Peter Seiler was supported by the U.S. Office of Naval Research (ONR) under Grant N00014-18-1-2209.

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

# Supplementary Material

## A  More on Problem Formulation

### A.1  Derivation of the cost function in frequency domain

The goal of the structured $\mathcal{H}_\infty$ synthesis with static output feedback is to find the optimal controller gain $K^*$ to solve the following minimax problem:

$$\min_{K \in \mathcal{K}} \max_{\mathbf{w}: \|\mathbf{w}\| \leq 1} \sum_{t=0}^{\infty} x_t^{\mathsf{T}} (Q + C^{\mathsf{T}} K^{\mathsf{T}} RKC) x_t, \tag{A.1}$$

Now if we define $z_t = (Q + C^{\mathsf{T}} K^{\mathsf{T}} RKC)^{\frac{1}{2}} x_t$, we have $\|z_t\|^2 = x_t^{\mathsf{T}} (Q + C^{\mathsf{T}} K^{\mathsf{T}} RKC) x_t = x_t^{\mathsf{T}} Q x_t + u_t^{\mathsf{T}} R u_t$. Then the closed-loop LTI system (2) can be viewed as a linear operator $G_K$ mapping any disturbance sequence $\{w_t\}$ to another sequence $\{z_t\}$. If $K$ is stabilizing, i.e. $\rho(A - BKC) < 1$, then $G_K$ is bounded in the sense that it maps any $\ell_2$ sequence $\mathbf{w}$ to another sequence $\mathbf{z}$ in $\ell_2^{n_x}$. For any stabilizing $K$, the $\ell_2 \to \ell_2$ induced norm of $G_K$ can be defined as:

$$\|G_K\|_{2\to 2} := \sup_{0 \neq \|\mathbf{w}\| \leq 1} \frac{\|\mathbf{z}\|}{\|\mathbf{w}\|} \tag{A.2}$$

Since $G_K$ is a linear operator, it is straightforward to show

$$\|G_K\|_{2\to 2}^2 := \max_{\mathbf{w}: \|\mathbf{w}\| \leq 1} \sum_{t=0}^{\infty} x_t^{\mathsf{T}} (Q + C^{\mathsf{T}} K^{\mathsf{T}} RKC) x_t = \max_{\mathbf{w}: \|\mathbf{w}\| \leq 1} \sum_{t=0}^{\infty} (x_t^{\mathsf{T}} Q x_t + u_t^{\mathsf{T}} R u_t).$$

Therefore, the minimax optimization problem (A.1) can be rewritten as the policy optimization problem: $\min_{K \in \mathcal{K}} \|G_K\|_{2\to 2}^2$. We can just drop the square in the cost function and reformulate (A.1) as $\min_{K \in \mathcal{K}} \|G_K\|_{2\to 2}$. This is exactly the policy optimization formulation for structured $\mathcal{H}_\infty$ with static output-feedback control. In the frequency domain, $G_K$ can be viewed as a transfer function that maps signal $w_t$ to $z_t$. Applying the frequency-domain formula for the $\mathcal{H}_\infty$ norm, we can calculate $\|G_K\|_{2\to 2}$ as

$$\|G_K\|_{2\to 2} = \sup_{\omega \in [0, 2\pi]} \sigma_{\max} \big( (Q + C^{\mathsf{T}} K^{\mathsf{T}} RKC)^{\frac{1}{2}} (e^{j\omega} I - A + BK)^{-1} \big), \tag{A.3}$$

which is the same as we defined in (5).

### A.2  LTI system with measurement noise

In this section, we extend the LTI system (1) considered in the main paper to the one with measurement noise as below:

$$x_{t+1} = Ax_t + Bu_t + w_t \quad x_0 = 0, \tag{A.4}$$
$$y_t = Cx_t + n_{t-1},$$

with $n_{-1} = 0$. We denote $\mathbf{n} := \{n_{-1}, n_0, \cdots\}$. We will show that the problem with measurement noise falls within our framework via a proper state augmentation trick. In particular, it is obvious that (A.4) can be rewritten as:

$$\begin{bmatrix} x_{t+1} \\ n_t \end{bmatrix} = \begin{bmatrix} A & 0 \\ 0 & 0 \end{bmatrix} \begin{bmatrix} x_t \\ n_{t-1} \end{bmatrix} + \begin{bmatrix} B \\ 0 \end{bmatrix} u_t + \begin{bmatrix} w_t \\ n_t \end{bmatrix}$$

$$y_t = \begin{bmatrix} C & I \end{bmatrix} \begin{bmatrix} x_t \\ n_{t-1} \end{bmatrix}.$$

In this setup, the control inputs become

$$u_t = -Ky_t = -KCx_t - Kn_{t-1}$$

and the closed-loop system can be represented as

$$\begin{bmatrix} x_{t+1} \\ n_t \end{bmatrix} = \begin{bmatrix} A - BKC & -BK \\ 0 & 0 \end{bmatrix} \begin{bmatrix} x_t \\ n_{t-1} \end{bmatrix} + \begin{bmatrix} w_t \\ n_t \end{bmatrix}.$$

Then, the min-max cost function can be written as

$$J(K) = \max_{\begin{bmatrix} \mathbf{w} \\ \mathbf{n} \end{bmatrix} : \left\| \begin{bmatrix} \mathbf{w} \\ \mathbf{n} \end{bmatrix} \right\| \leq 1} \sum_{t=0}^{\infty} x_t^\mathsf{T} Q x_t + u_t R u_t$$

$$= \max_{\begin{bmatrix} \mathbf{w} \\ \mathbf{n} \end{bmatrix} : \left\| \begin{bmatrix} \mathbf{w} \\ \mathbf{n} \end{bmatrix} \right\| \leq 1} \sum_{t=0}^{\infty} \begin{bmatrix} x_t \\ n_{t-1} \end{bmatrix}^\mathsf{T} \begin{bmatrix} Q & 0 \\ 0 & 0 \end{bmatrix} \begin{bmatrix} x_t \\ n_{t-1} \end{bmatrix} + (KCx_t + Kn_{t-1})^\mathsf{T} R(KCx_t + Kn_{t-1})$$

$$= \max_{\begin{bmatrix} \mathbf{w} \\ \mathbf{n} \end{bmatrix} : \left\| \begin{bmatrix} \mathbf{w} \\ \mathbf{n} \end{bmatrix} \right\| \leq 1} \sum_{t=0}^{\infty} \begin{bmatrix} x_t \\ n_{t-1} \end{bmatrix}^\mathsf{T} \begin{bmatrix} Q + C^\mathsf{T} K^\mathsf{T} RKC & C^\mathsf{T} K^\mathsf{T} RK \\ K^\mathsf{T} RKC & K^\mathsf{T} RK \end{bmatrix} \begin{bmatrix} x_t \\ n_{t-1} \end{bmatrix}.$$

If we define $z_t = \begin{bmatrix} Q + C^\mathsf{T} K^\mathsf{T} RKC & C^\mathsf{T} K^\mathsf{T} RK \\ K^\mathsf{T} RKC & K^\mathsf{T} RK \end{bmatrix}^{\frac{1}{2}} \begin{bmatrix} x_t \\ n_{t-1} \end{bmatrix}$, in frequency domain, we can rewrite the above cost function as

$$J(K) = \sup_{\omega \in [0, 2\pi]} \sigma_{\max} \left( \begin{bmatrix} Q + C^\mathsf{T} K^\mathsf{T} RKC & C^\mathsf{T} K^\mathsf{T} RK \\ K^\mathsf{T} RKC & K^\mathsf{T} RK \end{bmatrix}^{\frac{1}{2}} \left( e^{j\omega} I - \begin{bmatrix} A - BKC & -BK \\ 0 & 0 \end{bmatrix} \right)^{-1} \right).$$
(A.5)

We can slightly modify our proof of Lemma 3.2 to show that (A.5) remains coercive. Therefore, all the results presented in the paper follow.

## B   Detailed Proofs

### B.1   Proof of Lemma 3.2

Suppose we have a sequence $\{K^l\}$ satisfying $\|K^l\|_F \to +\infty$. Let $\mathbf{w}^l = \{w_0^l, 0, 0, \cdots\}$ with $\|w_0^l\| = 1$ such that $\sigma_{\max}(K^l C) = \|K^l C w_0^l\|$. Then we have:

$$J(K^l) = \max_{\mathbf{w}^l : \|\mathbf{w}^l\| \leq 1} \sum_{t=0}^{\infty} x_t^\mathsf{T} (Q + (K^l C)^\mathsf{T} RK^l C) x_t$$

$$\geq_{(i)} w_0^{l\,\mathsf{T}} (Q + (K^l C)^\mathsf{T} RK^l C) w_0^l$$

$$\geq_{(ii)} \lambda_{\min}(R) \|K^l C w_0^l\|^2$$

$$\geq \lambda_{\min}(R) \lambda_{\min}^{\frac{1}{2}}(CC^\mathsf{T}) \sigma_{\max}(K^l)$$

where inequality $(i)$ holds since we plugged into a specific $\mathbf{w}^l$ over the $\max$ operation and the matrix $Q + (K^l C)^\mathsf{T} RK^l C$ is positive definite. Inequality $(ii)$ uses the fact that $R \geq \lambda_{\min}(R)I$, where $\lambda_{\min}(R)$ is the minimum eigenvalue of $R$. Since $\|K^l\|_F \to +\infty$ by equivalence of norms $\sigma_{\max}(K^l) \to +\infty$, and knowing $C$ is a full row rank matrix guarantees that $\lambda_{\min}(CC^\mathsf{T}) > 0$, therefore $J(K^l) \to +\infty$ as $\|K^l\|_F \to +\infty$.

Next, we assume $K^l \to K$ where $K$ is on the boundary $\partial \mathcal{K}$. Clearly we have $\rho(A - BKC) = 1$. We will use a frequency-domain argument to prove $J(K^l) \to +\infty$. Since $\rho(A - BKC) = 1$, there exists some $\omega_0$ such that the matrix $(e^{j\omega_0} I - A + BKC)$ becomes singular. Obviously, for the same $\omega_0$, the matrix $(e^{j\omega_0} I - A + BKC)$ is also singular. Therefore, we have:

$$J(K^l) = \sup_{\omega \in [0, 2\pi]} \sigma_{\max} \left( (Q + (K^l C)^\mathsf{T} RK^l C)^{1/2} (e^{j\omega} I - A + BK^l C)^{-1} \right)$$

$$\geq \sup_{\omega \in [0, 2\pi]} \sigma_{\min} \left( (Q + (K^l C)^\mathsf{T} RK^l C)^{1/2} \right) \sigma_{\max} \left( (e^{j\omega} I - A + BK^l C)^{-1} \right)$$

$$\geq \lambda_{\min}^{1/2}(Q) \sigma_{\max} \left( (e^{j\omega_0} I - A + BK^l C)^{-1} \right).$$

Clearly, the above argument relies on the fact that $Q$ is positive definite and $Q \geq \lambda_{\min}(Q)I$. Notice that we have $\rho(A - BK^l C) < 1$ for each $l$, and hence we have $\sigma_{\min} \left( (e^{j\omega_0} I - A + BK^l C) \right) > 0$,

i.e. the smallest singular values of $(e^{j\omega_0}I - A + BK^lC)$ are strictly positive for all $l$. By the continuity of the $\sigma_{\min}(\cdot)$, we must have $\sigma_{\min}\left((e^{j\omega_0}I - A + BK^lC)\right) \to 0$ as $\lim_{l\to\infty} K^l \in \partial\mathcal{K}$. Hence we have $\sigma_{\max}\left((e^{j\omega_0}I - A + BK^lC)^{-1}\right) \to +\infty$ as $l \to \infty$. Therefore, we have $J(K^l) \to +\infty$ as $K^l \to K \in \partial\mathcal{K}$. This completes the proof.

## B.2   Proof of Lemma 3.3

We first prove Lemma 3.3. Since $J$ is continuous, for any $\gamma > J^*$, we know $\mathcal{S}_\gamma = \{K \in \mathcal{K} : J(K) \leq \gamma\}$ is a closed set. It remains to show $\mathcal{S}_\gamma$ is bounded. Suppose that $\mathcal{S}_\gamma$ is unbounded. Then there exists a sequence $\{K^l\}_{l=1}^\infty \subset \mathcal{S}$ such that $\|K^l\|_2 \to +\infty$ as $l \to \infty$. But by coerciveness of $J(K)$, we must have $J(K^l) \to +\infty$ as well, which contradicts that $J(K^l) \leq \gamma$ for all $l$. Hence $\mathcal{S}_\gamma$ is bounded. Therefore, $\mathcal{S}_\gamma$ is compact.

## B.3   Proof of Theorem 3.6

We first use induction to show that the iterative method (8) stays in $\mathcal{K}$ for all $t$. Since $K^0 \in \mathcal{S}^0$ and $\delta \leq \Delta_1$, we know that $J_\delta(K^0)$ is well defined. In addition, we have:

$$\|K^1 - K^0\|_F = \eta\|\nabla J_\delta(K^0)\|_F \leq \eta L_1 \leq \xi, \tag{B.1}$$

where we use the fact that $J_\delta$ is $L_1$-Lipschitz on the sublevel set $S^1$ and $\eta \leq \frac{\xi}{L_1}$. Therefore we have $K^1 \in \mathcal{S}^1$ by Remark 3.5 and $J_\delta(K^1)$ is well defined. Since $J_\delta$ is $L_1$-Lipschitz with the $\frac{cL_1\sqrt{d}}{\delta}$-Lipschtiz gradient, we have:

$$J_\delta(K^1) - J_\delta(K^0) \leq \langle \nabla J_\delta(K^0), K^1 - K^0 \rangle + \eta^2 \frac{cL_1\sqrt{d}}{2\delta}\|\nabla J_\delta(K^0)\|_F^2$$

$$= (\eta^2 \frac{cL_1\sqrt{d}}{2\delta} - \eta)\|\nabla J_\delta(K^0)\|_F^2$$

$$\leq -\frac{\eta}{2}\|\nabla J_\delta(K^0)\|_F^2,$$

where the last inequality holds since we have $0 < \eta = \min\{\frac{\xi}{L_1}, \frac{\delta}{cL_1\sqrt{d}}\}$. Therefore, we have $J_\delta(K^1) \leq J_\delta(K^0)$. This implies:

$$J(K^1) \leq J_\delta(K^1) + \delta L_1 \leq J_\delta(K^0) + \delta L_1 \leq J(K^0) + 2\delta L_1 \leq 50J(K^0). \tag{B.2}$$

Hence we have $K^1 \in \mathcal{S}^0$. Repeating this argument, leads to the fact that $K^t \in \mathcal{S}^0$ for all $t$. Thus:

$$J_\delta(K^{t+1}) \leq J_\delta(K^t) + \langle \nabla J_\delta(K^t), K^{t+1} - K^t \rangle + \eta^2 \frac{cL_1\sqrt{d}}{2\delta}\|\nabla J_\delta(K^t)\|_F^2$$

$$\leq J_\delta(K^t) - \frac{\eta}{2}\|\nabla J_\delta(K^t)\|_F^2.$$

Rearranging terms of the above inequality yields:

$$\frac{\eta}{2}\|\nabla J_\delta(K^t)\|_F^2 \leq J_\delta(K^t) - J_\delta(K^{t+1}).$$

Summing up the above inequality over $t = 0, 1, \cdots, T-1$ gives:

$$\frac{\eta}{2T}\sum_{t=0}^{T-1}\|\nabla J_\delta(K^t)\|_F^2 \leq \frac{J_\delta(K^0) - J_\delta(K^T)}{T} \leq \frac{J_\delta(K^0) - J^*}{T},$$

where we use the inequality $J_\delta(K^T) \geq J^*$ by definition of $J_\delta(K)$. Therefore, we have:

$$\min_{t=0,1,\cdots,T-1}\|\nabla J_\delta(K^t)\|_F^2 \leq \frac{1}{T}\sum_{t=0}^{T-1}\|\nabla J_\delta(K^t)\|_F^2 \leq \frac{2(J_\delta(K^0) - J^*)}{\eta T},$$

Therefore, we have:

$$\min_{t=0,1,\cdots,T-1}\|\nabla J_\delta(K^t)\|_F \leq \sqrt{\frac{2(J_\delta(K^0) - J^*)}{\eta T}}$$

and the sample complexity result follows. This completes the proof.

## B.4  Proof of Theorem 3.7

We first provide several technical lemmas which will be useful for proving Theorem 3.7. To this end, let $\mathcal{F}_t$ denote the filtration $\sigma(K^s, s \leq t)$ for each $t = 1, 2, \cdots, T$.

**Lemma B.1.** *Suppose $J_\delta(K) \leq 49J(K^0)$ and $\delta \leq \frac{J(K^0)}{L_1}$. Then we have $K \in \mathcal{S}^0$.*

*Proof.* Since $J_\delta(K)$ is well defined, we must have $K \in \mathcal{K}$ by definition of $J_\delta$. Then Proposition 2.3 (i) implies that $J(K) \leq J_\delta(K) + \delta L_1 \leq 49J(K^0) + \frac{J(K^0)}{L_1}L_1 = 50J(K^0)$. Hence we have $K \in \mathcal{S}^0$. $\qquad\square$

**Lemma B.2.** *Suppose that $J(K)$ is $L_1$-Lipschitz on the sublevel set $\mathcal{S}^1$ and let $\{g^t\}_{t=0}^{T-1}$ and $\{K^t\}_{t=0}^{T-1}$ be generated by Algorithm 1 such that $\{K^t\}_{t=0}^{T-1}$ are feasible and $\{J_\delta(K^t)\}_{t=0}^{T-1}$ are well defined. Then, we have*

$$\mathbb{E}[g^t \mid \mathcal{F}_t] = \nabla J_\delta(K^t), \tag{B.3}$$

$$\mathbb{E}[\|g^t\|_F^2 \mid \mathcal{F}_t] \leq 16\sqrt{2\pi}dL_1^2. \tag{B.4}$$

*Proof.* The proof can be found in [46, Lemma D.1]. We omit it here. $\qquad\square$

**Lemma B.3.** *$J(K)$ is $L_0$-Lipschitz on the sublevel set $\mathcal{S}^0$, let $\eta \leq \frac{2\delta\xi}{d(100J(K^0)-J^*)}$ and $\delta \leq \min\{\Delta_1, \Delta\}$, then as long as $K^t \in \mathcal{S}^0$, we will have $K^{t+1} \in \mathcal{S}^1$ and*

$$\mathbb{E}[J_\delta(K^{t+1}) \mid \mathcal{F}_t] \leq J_\delta(K^t) - \eta\|\nabla J_\delta(K^t)\|_F^2 + \eta Z, \tag{B.5}$$

*where $Z = \eta(8\sqrt{2\pi})cd^{3/2}L_1^3\delta^{-1}$.*

*Proof.* Since $\|W^t\|_F = 1$, we have:

$$
\begin{aligned}
\|K^{t+1} - K^t\|_F &= \eta\|\frac{d}{2\delta}(J(K^t + \delta W^t) - J(K^t - \delta W^t))W^t\|_F \\
&\leq \frac{\eta d}{2\delta}|(J(K^t + \delta W^t) - J(K^t - \delta W^t)| \\
&\leq \frac{\eta d}{2\delta} \cdot (100J(K^0) - J^*) \\
&\leq \xi,
\end{aligned}
$$

where the second inequality holds since we have $K^t \pm \delta W^t \in \mathcal{S}^1$ when $\delta \leq \Delta$. This implies that $K^{t+1} \in \mathcal{S}^1$ by Remark 3.5. In addition, $K^{t+1} \in \mathcal{S}^1$ and $\delta \leq \Delta_1$ ensures that $J_\delta(K^{t+1})$ is well defined. By Proposition 2.3 (ii), we know that $J_\delta(K)$ is differentiable and $L_1$-Lipschitz with $\frac{cL_1\sqrt{d}}{\delta}$-Lipschitz gradient on the sublevel set $\mathcal{S}^1$. Then we have:

$$J_\delta(K^{t+1}) \leq J_\delta(K^t) - \eta\langle\nabla J_\delta(K^t), g^t\rangle + \frac{c\eta^2 L_1\sqrt{d}}{2\delta}\|g^t\|_F^2. \tag{B.6}$$

Taking the expectation on both sides conditioned on $\mathcal{F}_t$ and using Lemma B.2 yields

$$
\begin{aligned}
\mathbb{E}[J_\delta(K^{t+1}) \mid \mathcal{F}_t] &\leq J_\delta(K^t) - \eta\langle\nabla J_\delta(K^t), \mathbb{E}[g^t \mid \mathcal{F}_t]\rangle + \frac{c\eta^2 L_1\sqrt{d}}{2\delta}\mathbb{E}[\|g^t\|_F^2 \mid \mathcal{F}_t] \\
&\leq J_\delta(K^t) - \eta\|\nabla J_\delta(K^t)\|_F^2 + \frac{c\eta^2 L_1\sqrt{d}}{2\delta}16\sqrt{2\pi}dL_1^2 \\
&= J_\delta(K^t) - \eta\|\nabla J_\delta(K^t)\|_F^2 + \eta Z,
\end{aligned}
$$

with $Z = \eta(8\sqrt{2\pi})cd^{3/2}L_1^3\delta^{-1}$. This completes the proof. $\qquad\square$

Now we are ready to prove the Theorem 3.7. We will first prove Statement 1 in Theorem 3.7: all the generated controllers are within the feasible set with high probability. Then we will show Statement 2: the Algorithm 1 returns a $(\delta, \epsilon)$-stationary point with high probability.

**Proof of Statement 1** We first define a stopping time $\tau$ as below:

$$\tau := \min\{t \in \{0, 1, 2, \cdots, T-1\} : J_\delta(K^t) > 49J(K^0)\}. \tag{B.7}$$

Based on Lemma B.1, it can be seen that as long as $\tau \geq T-1$, the iterates $\{K^t\}_{t=0}^{T-1}$ generated by Algorithm 1 are feasible. Therefore our goal becomes bounding the probability $\Pr(\tau \leq T-1)$. To this end, we can define a nonnegative supermartingale $Y(t)$ as below

$$Y(t) := J_\delta(K^{\min\{t, \tau\}}) + (T-t)\eta Z. \tag{B.8}$$

To show it is a supermartingale, noticing that we have

$$\mathbb{E}[Y(t+1) \mid \mathcal{F}_t] = \mathbb{E}[J_\delta(K^\tau)\mathbb{1}_{\tau \leq t} \mid \mathcal{F}_t] + \mathbb{E}[J_\delta(K^{t+1})\mathbb{1}_{\tau > t} \mid \mathcal{F}_t] + (T-t-1)\eta Z$$

$$=_{(i)} J_\delta(K^\tau)\mathbb{1}_{\tau \leq t} + \mathbb{E}[J_\delta(K^{t+1})\mathbb{1}_{\tau > t} \mid \mathcal{F}_t] + (T-t-1)\eta Z$$

$$\leq_{(ii)} J_\delta(K^\tau)\mathbb{1}_{\tau \leq t} + J_\delta(K^t)\mathbb{1}_{\tau > t} - \eta\|\nabla J_\delta(K^t)\|_F^2 + \eta Z + (T-t-1)\eta Z$$

$$\leq J_\delta(K^{\min\{t, \tau\}}) + (T-t)\eta Z = Y(t),$$

where in equality (i), $J_\delta(K^{t+1})$ is well defined by the definition of the $\tau$; in particular, we have $J_\delta(K^t) \leq 49J(K^0)$, hence $K^t \in \mathcal{S}^0$ by Lemma B.1 and we can apply Lemma B.3 to obtain inequality (ii). Then Doob's maximal inequality for super-martingales gives

$$\Pr(\tau \leq T-1) \leq \Pr(\max_{t=0,1,\cdots,T-1} Y(t) > 49J(K^0))$$

$$\leq \frac{\mathbb{E}[Y(0)]}{49J(K^0)} = \frac{J_\delta(K^0) + T\eta Z}{49J(K^0)} \leq_{(i)} \frac{J(K^0) + \delta L_1}{49J(K^0)} + \frac{T\eta Z}{49J(K^0)}$$

$$\leq_{(ii)} \frac{2}{49} + \frac{T\eta Z}{49J(K^0)},$$

where inequality (i) uses Lemma 2.3 and (ii) is true since $\delta \leq \frac{J(K^0)}{L_1}$. For sufficiently small $\epsilon$, we have $\eta = \frac{\delta\epsilon^2}{3500cd^{3/2}L_1^3}$, then one can verify that $T\eta Z = \frac{4\sqrt{2}\pi\upsilon}{25}J(K^0)$. Therefore, we have

$$\Pr(\tau \leq T-1) \leq \frac{2}{49} + \frac{T\eta Z}{49J(K^0)} \leq \frac{2}{49} + \frac{4\sqrt{2}\pi\upsilon}{1225} \leq \frac{1}{24} + \frac{\upsilon}{100} \leq 0.05 + 0.01\upsilon.$$

This implies that all the iterates $K^t$ are stabilizing with probability at least $1 - (0.05 + 0.01\upsilon) = 0.95 - 0.01\upsilon$. This completes the proof for Statement 1.

**Proof of Statement 2** We first show that we can extend (B.5) as

$$\mathbb{E}[J_\delta(K^{t+1})\mathbb{1}_{\tau > t+1} \mid \mathcal{F}_t] \leq J_\delta(K^t)\mathbb{1}_{\tau > t} - \eta\|\nabla J_\delta(K^t)\|_F^2\mathbb{1}_{\tau > t} + \eta Z, \tag{B.9}$$

If $\tau > t$, then we know $J_\delta(K^t) \leq 49J(K^0)$, hence $J(K^t) \leq J_\delta(K^t) + \delta L_1 \leq 50J(K^0)$, we have $K^t \in \mathcal{S}^0$ and $K^{t+1} \in \mathcal{S}^1$. Therefore, we can apply Lemma B.3 to show that:

$$\mathbb{E}[J_\delta(K^{t+1})\mathbb{1}_{\tau > t+1} \mid \mathcal{F}_t] \leq \mathbb{E}[J_\delta(K^{t+1}) \mid \mathcal{F}_t]$$

$$\leq J_\delta(K^t) - \eta\|\nabla J_\delta(K^t)\|_F^2 + \eta Z$$

$$= J_\delta(K^t)\mathbb{1}_{\tau > t} - \eta\|\nabla J_\delta(K^t)\|_F^2\mathbb{1}_{\tau > t} + \eta Z.$$

On the other hand, if $\tau \leq t$, we have

$$\mathbb{E}[J_\delta(K^{t+1})\mathbb{1}_{\tau > t+1} \mid \mathcal{F}_t] = 0 \leq J_\delta(K^t)\mathbb{1}_{\tau > t} - \eta\|\nabla J_\delta(K^t)\|_F^2\mathbb{1}_{\tau > t} + \eta Z \tag{B.10}$$

since $Z \geq 0$. Therefore, (B.9) holds. Taking the expectation and rearranging the terms of (B.9) gives:

$$\mathbb{E}[\|\nabla J_\delta(K^t)\|_F^2\mathbb{1}_{\tau > t}] \leq \frac{\mathbb{E}[J_\delta(K^t)\mathbb{1}_{\tau > t}] - \mathbb{E}[J_\delta(K^{t+1})\mathbb{1}_{\tau > t+1}]}{\eta} + Z. \tag{B.11}$$

Summing up the above inequality over $t = 0, 1, \cdots, T-1$ yields

$$\frac{1}{T}\sum_{t=0}^{T-1} \mathbb{E}[\|\nabla J_\delta(K^t)\|_F^2\mathbb{1}_{\tau > T-1}] \leq \frac{1}{T}\sum_{t=0}^{T-1} \mathbb{E}[\|\nabla J_\delta(K^t)\|_F^2\mathbb{1}_{\tau > t}] \leq \frac{J_\delta(K^0) - \mathbb{E}[J_\delta(K^T)\mathbb{1}_{\tau > T}]}{\eta T} + Z.$$

Since the cost function $J(K)$ is nonnegative, we have $\mathbb{E}[J_\delta(K^T)\mathbb{1}_{\tau>T}] \geq 0$ and:

$$J_\delta(K^0) - \mathbb{E}[J_\delta(K^T)\mathbb{1}_{\tau>T}] \leq J_\delta(K^0) \leq J(K^0) + \delta L_1 = 2J(K^0).$$

Hence we conclude that

$$\frac{1}{T}\sum_{t=0}^{T-1}\mathbb{E}[\|\nabla J_\delta(K^t)\|_F^2 \mathbb{1}_{\tau>T-1}] \leq \frac{2J(K^0)}{\eta T} + Z.$$

By the choice of $\eta$ and $T$, the above inequality becomes:

$$\frac{1}{T}\sum_{t=0}^{T-1}\mathbb{E}[\|\nabla J_\delta(K^t)\|_F^2 \mathbb{1}_{\tau>T-1}] \leq \frac{2J(K^0)}{\eta T} + Z \leq (\frac{5}{\upsilon}+1)Z \leq (\frac{5}{\upsilon}+1)\frac{8\sqrt{2\pi}}{3500}\epsilon^2.$$

Since the random count $R \in \{0, 1, 2, \cdots, T-1\}$ is uniformly sampled, we have

$$\mathbb{E}[\|\nabla J_\delta(K^R)\|_F^2 \mathbb{1}_{\tau>T-1}] = \frac{1}{T}\sum_{t=0}^{T-1}\mathbb{E}[\|\nabla J_\delta(K^t)\|_F^2 \mathbb{1}_{\tau>T-1}] \leq (\frac{5}{\upsilon}+1)\frac{8\sqrt{2\pi}}{3500}\epsilon^2,$$

which implies

$$\mathbb{E}[\|\nabla J_\delta(K^R)\|_F \mathbb{1}_{\tau>T-1}] \leq (0.17\upsilon^{-\frac{1}{2}} + 0.08)\epsilon.$$

Therefore, we have

$$\begin{aligned}
\Pr(\|\nabla J_\delta(K^R)\|_F \geq \epsilon) &= \Pr(\|\nabla J_\delta(K^R)\|_F \geq \epsilon, \tau > T-1) + \Pr(\|\nabla J_\delta(K^R)\|_F \geq \epsilon, \tau \leq T-1)\\
&\leq \Pr(\|\nabla J_\delta(K^R)\|_F \geq \epsilon, \tau > T-1) + \Pr(\tau \leq T-1)\\
&\leq \frac{1}{\epsilon}\mathbb{E}[\|\nabla J_\delta(K^R)\|_F \mathbb{1}_{\tau>T-1}] + \Pr(\tau \leq T-1)\\
&\leq 0.17\upsilon^{-\frac{1}{2}} + 0.08 + \frac{1}{24} + \frac{\upsilon}{100}\\
&\leq 0.17\upsilon^{-\frac{1}{2}} + 0.13 + 0.01\upsilon,
\end{aligned}$$

where the second inequality uses the Markov's inequality. By Proposition 2.3 (iii), we have $\nabla J_\delta(K^R) \in \partial_\delta J(K^R)$. This implies that

$$\Pr(\min\{\|H\|_F : H \in \partial_\delta J(K^R) \geq \epsilon\}) \leq \Pr(\|\nabla J_\delta(K^R)\|_F \geq \epsilon) \leq 0.17\upsilon^{-\frac{1}{2}} + 0.13 + 0.01\upsilon.$$

Therefore our algorithm can find a $(\delta, \epsilon)$-stationary point with probability at least $1 - (0.17\upsilon^{-\frac{1}{2}} + 0.13 + 0.01\upsilon) = 0.87 - 0.17\upsilon^{-\frac{1}{2}} - 0.01\upsilon$. This completes the proof for Statement 2.

*Remark* B.4. It is worth mentioning that the choice of sublevels in sets in (9) is WLOG. Tuning these two numbers entails trade-offs. On the one hand, a larger sublevel in set $\mathcal{S}^1$ yields a higher probability as observed from the proofs of Theorem 3.7 where we are bounding the term $\Pr(\tau \leq T-1)$ as an example. On the other hand, a larger sublevel set also results in a larger Lipschitz constant $L_1$ and a smaller smooth radius $\delta$, thereby worsening the complexity bounds.

## B.5 Proof of Theorem 4.1

We first state several technical lemmas which will be useful in proving of Theorem 4.1.

**Lemma B.5.** *Let $P(z) = \sum_{k=0}^{\infty} P_k z^{-k}$ be a stable, discrete-time MIMO LTI system, where $P = \{P_k \in \mathbb{C}^{n_x \times n_x}\}_{k \in \mathbb{Z}_+}$ is the impulse response of $P(z)$ and $T(P) = (P_{i-j})_{i,j=0}^{\infty,i}$ denote the Toeplitz operator associated with system $P(z)$. Suppose for any given Euclidean unit vectors $y \in \mathbb{C}^{n_x}$ and $x \in \mathbb{C}^{n_x}$, we define a scalar sequence $g = \{g_k = y^* P_k x\}_{k \in \mathbb{Z}_+}$. Let $G(z) = \sum_{k=0}^{\infty} g_k z^{-k}$ denote the z-transform of the impulse response $g$ and $T(g) = (g_{i-j})_{i,j=0}^{\infty,i}$ be the corresponding Toeplitz matrix. For any given Euclidean unit vectors $x$ and $y$, we can show:*

$$\|T_N(g)\| \leq \|T_N(P)\| \tag{B.12}$$

*where $T_N(P)$ and $T_N(g)$ are $Nn_x \times Nn_x$ and $N \times N$ upper-left submatrix of $T(P)$ and $T(g)$, respectively, $\|\cdot\|$ is the spectral norm of a given matrix.*

*Proof.* Using the definition of $g_k = y^* P_k x$, we can write $T_N(g)$ as:

$$T_N(g) = \begin{bmatrix} g_0 & 0 & \cdots & 0 \\ g_1 & g_0 & \cdots & 0 \\ \vdots & \vdots & \ddots & \vdots \\ g_{N-1} & g_{N-2} & \cdots & g_0 \end{bmatrix} = \begin{bmatrix} y^* P_0 x & 0 & \cdots & 0 \\ y^* P_1 x & y^* P_0 x & \cdots & 0 \\ \vdots & \vdots & \ddots & \vdots \\ y^* P_{N-1} x & y^* P_{N-2} x & \cdots & y^* P_0 x \end{bmatrix}$$

$$= \underbrace{\begin{bmatrix} y^* & \cdots & 0 \\ \vdots & \ddots & \vdots \\ 0 & \cdots & y^* \end{bmatrix}}_{Y^*} \underbrace{\begin{bmatrix} P_0 & 0 & \cdots & 0 \\ P_1 & P_0 & \cdots & 0 \\ \vdots & \vdots & \ddots & \vdots \\ P_{N-1} & P_{N-2} & \cdots & P_0 \end{bmatrix}}_{T_N(P)} \underbrace{\begin{bmatrix} x & \cdots & 0 \\ \vdots & \ddots & \vdots \\ 0 & \cdots & x \end{bmatrix}}_{X}$$

(B.13)

Since $\|x\| = \|y\| = 1$, we have $\sigma_{\max}(Y^*) = \sigma_{\max}(X) = 1$.

$$\begin{aligned} \|T_N(g)\| = \sigma_{\max}(T_N(g)) &= \sigma_{\max}(Y^* T_N(P) X) \\ &\leq \sigma_{\max}(Y^*) \sigma_{\max}(T_N(P)) \sigma_{\max}(X) \\ &\leq \sigma_{\max}(T_N(P)) = \|T_N(P)\| \end{aligned}$$

(B.14)

which completes the proof. $\qquad\square$

**Corollary B.6.** *Let $P(z)$ and $G(z)$ be defined as lemma B.5, fix a $\gamma \in (\iota, 1)$ and define the systems $P^\gamma(z) := \gamma P(\gamma z)$ and $G^\gamma(z) := \gamma G(\gamma z)$. We can show that:*

$$\|G^\gamma\|_\infty \leq \|P^\gamma\|_\infty$$

(B.15)

*Proof.* This result can be shown by applying lemma B.5 to $G^\gamma(z)$ and $P^\gamma(z)$ and considering this fact that $\|T_N(g^\gamma)\|$ and $\|T_N(P^\gamma)\|$ converge to $\|G^\gamma\|_\infty$ and $\|P^\gamma\|_\infty$ as $N \to \infty$, respectively. $\quad\square$

Now we are ready to extend [68] results of $\mathcal{H}_\infty$-norm approximation of SISO system using the corresponding truncated Toeplitz matrix to more general MIMO systems.

**Lemma B.7.** *Let $P(z) = \sum_{k=0}^\infty P_k z^{-k}$ be a stable, discrete-time MIMO LTI system with stability radius $\iota \in (0, 1)$. Fix a $\gamma \in (\iota, 1)$ and suppose that $\sigma_{max}(P_0) \leq D$. For all $N \geq 3$, we have that*

$$\|P(z)\|_\infty - \|T_N(P)\| \leq C_1 \frac{D\|P^\gamma(z)\|_\infty(1 - \gamma^2) + \|P^\gamma(z)\|_\infty^2 \gamma}{\|P(z)\|_\infty(1 - \gamma)^4} \frac{1}{N^2}$$

(B.16)

$$+ C_2 \frac{\|P^\gamma(z)\|_\infty^2}{\|P(z)\|_\infty(1 + \gamma)(1 - \gamma)^5} \frac{1}{N^3}$$

*where $\|P^\gamma(z)\|_\infty$ denotes the $\mathcal{H}_\infty$-norm of the system $P^\gamma(z) := \gamma P(\gamma z)$, $C_1 = 3\sqrt{2}\pi(2 + 3\pi^4)$, $C_2 = 9\sqrt{2}\pi^2$ are universal constants.*

*Proof.* We can write $\|P\|_\infty$ as follows

$$\|P\|_\infty = \max_{\omega \in [0, 2\pi]} \sigma_{\max}(P(e^{i\omega})) = \sigma_{\max}(P(e^{i\omega_{\text{opt}}}))$$

(B.17)

This means that there exists Euclidean unit vectors $x$ and $y$ such that $\|P\|_\infty = y^* P(e^{i\omega_{\text{opt}}}) x$. Now define a SISO LTI system $G(z) = y^* P(z) x$, which has the impulse response $g = \{g_k = y^* P_k x\}_{k \in \mathbb{Z}_+}$. Since $P(z)$ is a stable function, so is $G(z)$ and we have $\|P\|_\infty = \|G\|_\infty$. Additionally, regarding the stability radius of $G$, it can be derived that, $\iota_g \leq \iota$ and $\gamma \in (\iota_g, 1)$. Consequently:

$$\|g_0\| = \|y^* P_0 x\| \leq \sigma_{max}(P_0) \leq D$$

(B.18)

Now we invoke Theorem 4.1 from [68]; for a stable, SISO LTI system $G(z) = \sum_{k=0}^\infty g_k z^{-k}$, with a fix $\gamma \in (\iota_g, 1)$ and $\|g_0\| \leq D$, we have:

$$\|G\|_\infty - \|T_N(g)\| \leq C_1 \frac{D\|G^\gamma\|_\infty(1 - \gamma^2) + \|G^\gamma\|_\infty^2 \gamma}{\|G\|_\infty(1 - \gamma)^4} \frac{1}{N^2} + C_2 \frac{\|G^\gamma\|_\infty^2}{\|G\|_\infty(1 + \gamma)(1 - \gamma)^5} \frac{1}{N^3}$$

From lemma B.5 and corollary B.6, we know that $\|T_N(g)\| \leq \|T_N(P)\|$ and $\|G^\gamma\|_\infty \leq \|P^\gamma\|_\infty$. Also using this fact that $\|G\|_\infty = \|P\|_\infty$, we have

$$\|\|P\|_\infty - \|T_N(P)\|\| \leq \zeta_1$$

(B.19)

where $\zeta_1 = C_1 \frac{D\|P^\gamma\|_\infty(1-\gamma^2)+\|P^\gamma\|_\infty^2 \gamma}{\|P\|_\infty(1-\gamma)^4} \frac{1}{N^2} + C_2 \frac{\|P^\gamma\|_\infty^2}{\|P\|_\infty(1+\gamma)(1-\gamma)^5} \frac{1}{N^3}$. $\qquad\square$

**Algorithm 2:** Power iteration method to find the maximum eigenvalue of a matrix

**Require:** Given a positive semi-definite matrix $M \in \mathcal{R}^{Nn_x \times Nn_x}$ and an initial vector $v^{(0)} \in \mathcal{R}^{Nn_x}$

**for** $i = 0, 1, \cdots, n$ **do**
   $z^{(i)} = Mv^{(i-1)}$.
   $v^{(i)} = z^{(i)}/\|z^{(i)}\|_2$.
   $\lambda^{(i)} = [v^{(i)}]^* Mv^{(i)}$.
**end for**
**Output:** $\lambda^{(n)}$.

**Lemma B.8.** *Given a positive semi-definite matrix $M \in \mathbb{R}^{Nn_x \times Nn_x}$ with spectral factorization*

$$V^* MV = diag(\lambda_1, ..., \lambda_{Nn_x})$$

*Where $V = [v_1|...|v_{Nn_x}]$ is orthogonal, $\lambda_1 \geq \lambda_2 \geq \cdots \geq \lambda_{Nnx} \geq 0$ and $\lambda_1 > 0$. Suppose we choose a $\psi$ such that $\lambda_1, \cdots, \lambda_j \in [\lambda_1 - \psi, \lambda_1]$. Let the vectors $v^{(i)}$ be specified by Algorithm 2 and define $\theta_0 \in [0, \pi/2)$ by $\cos(\theta_0) = |v_1^T v^{(0)}|$. For $i = 0, 1, ...$ we have*

$$|\lambda^{(i)} - \lambda_1| \leq \tan(\theta_0)^2 \left( \psi + \lambda_1 \left( \frac{\lambda_1 - \psi}{\lambda_1} \right)^{2i} \right) \tag{B.20}$$

*Proof.* The idea of our proof is based on the results in section 8.2.1 of [29]. Suppose $v^{(0)}$ has the eigenvector expansion $v^{(0)} = a_1 v_1 + a_2 v_2 + ... + a_{Nn_x} v_{Nn_x}$, then

$$|a_1| = |v_1^T v^{(0)}| = \cos(\theta_0) \neq 0$$
$$a_1^2 + a_2^2 + ... + a_{Nnx}^2 = 1$$

and

$$\lambda^{(i)} = [v^{(i)}]^T Mv^{(i)} = \frac{[v^{(0)}]^T M^{2i+1} v^{(0)}}{[v^{(0)}]^T M^{2i} v^{(0)}} = \frac{\sum_{k=1}^{Nn_x} a_k^2 \lambda_k^{2i+1}}{\sum_{k=1}^{Nn_x} a_k^2 \lambda_k^{2i}}$$

and so

$$|\lambda^{(i)} - \lambda_1| = \left| \frac{\sum_{k=2}^{Nn_x} a_k^2 \lambda_k^{2i}(\lambda_k - \lambda_1)}{\sum_{k=1}^{Nn_x} a_k^2 \lambda_k^{2i}} \right| \leq \left| \frac{\sum_{k=2}^{Nn_x} a_k^2 \lambda_k^{2i}(\lambda_k - \lambda_1)}{a_1^2 \lambda_1^{2i}} \right|$$

$$= \left| \frac{\sum_{k=2}^{j} a_k^2 \lambda_k^{2i}(\lambda_k - \lambda_1)}{a_1^2 \lambda_1^{2i}} \right| + \left| \frac{\sum_{k=j+1}^{Nn_x} a_k^2 \lambda_k^{2i}(\lambda_k - \lambda_1)}{a_1^2 \lambda_1^{2i}} \right|$$

$$\leq \psi \frac{1}{a_1^2} \sum_{k=2}^{j} a_k^2 \left( \frac{\lambda_k}{\lambda_1} \right)^{2i} + \frac{1}{a_1^2} \left| \sum_{k=j+1}^{Nn_x} a_k^2 (\lambda_k - \lambda_1) \left( \frac{\lambda_k}{\lambda_1} \right)^{2i} \right|$$

$$\leq \psi \frac{1}{a_1^2} \sum_{k=2}^{j} a_k^2 \left( \frac{\lambda_2}{\lambda_1} \right)^{2i} + \lambda_1 \frac{1}{a_1^2} \sum_{k=j+1}^{Nn_x} a_k^2 \left( \frac{\lambda_{j+1}}{\lambda_1} \right)^{2i}$$

$$\leq_{(i)} \tan(\theta_0)^2 \left( \psi \left( \frac{\lambda_2}{\lambda_1} \right)^{2i} + \lambda_1 \left( \frac{\lambda_{j+1}}{\lambda_1} \right)^{2i} \right)$$

$$\leq \tan(\theta_0)^2 \left( \psi + \lambda_1 \left( \frac{\lambda_1 - \psi}{\lambda_1} \right)^{2i} \right)$$

where the inequality $(i)$ follows from the fact that $\tan(\theta_0)^2 = \frac{1-a_1^2}{a_1^2}$ and this completes the proof. $\square$

**Corollary B.9.** *Let matrix $M$ be given as lemma B.8, we can show that for each iteration $i$, $\psi_{\min}^{(i)} = \lambda_1 \left( 1 - \frac{1}{2i - \sqrt[2i]{2i}} \right)$ minimize the upper bound in power iteration algorithm (B.20), and (B.20) can be written as:*

$$|\lambda^{(i)} - \lambda_1| \leq \tan(\theta_0)^2 \lambda_1 \left( 1 - \frac{1}{2i - \sqrt[2i]{2i}} \left( 1 - \frac{1}{2i} \right) \right) \tag{B.21}$$

*Proof.* Since the right hand side of inequality (B.20) is a convex function of $\psi$, we can minimize it by setting its derivative with respect to $\psi$ equal to zero. Define $f(\psi) = \tan(\theta_0)^2 \left( \psi + \lambda_1 \left( \frac{\lambda_1 - \psi}{\lambda_1} \right)^{2i} \right)$, we have:

$$f'(\psi) = \tan(\theta_0)^2 \left( 1 - 2i \left( \frac{\lambda_1 - \psi}{\lambda_1} \right)^{2i-1} \right) = 0$$

It is straightforward to show that $\psi_{\min}^{(i)} = \lambda_1 \left( 1 - \frac{1}{2i - \sqrt[2i]{2i}} \right)$ makes $f'(\psi) = 0$ and substitute $\psi_{\min}^{(i)}$ in $f(\psi)$, gives $f(\psi_{\min}^{(i)}) = \tan(\theta_0)^2 \lambda_1 \left( 1 - \frac{1}{2i - \sqrt[2i]{2i}} \left( 1 - \frac{1}{2i} \right) \right)$, which completes the proof. $\square$

Now we are ready to prove Theorem 4.1. As mentioned before, cost function $J(K)$ is equal to $\mathcal{H}_\infty$-norm of $P(z)$, $\|P\|_\infty$. From lemma B.7, We can find the bound $\zeta_1(K)$ for $|J(K) - \|T_N(P)\|| \leq \zeta_1(K)$. Also, using Algorithm 2 we can obtain the model-free cost function $J(K)$ as $\hat{J}(K) := \sqrt{\lambda^{(n)}}$. It is also well-known that $\sqrt{\lambda_1} \leq \|P\|_\infty$ and $\sqrt{\lambda_1} \to \|P\|_\infty$ as $N \to \infty$. Now invoke lemma B.8, corollary B.9 and use $M := T_N(P)^* T_N(P)$, we have

$$|\hat{J}(K)^2 - \|T_N(P)\|^2| \leq \tan(\theta_0)^2 \lambda_1 \left( 1 - \frac{1}{2n - \sqrt[2n]{2n}} \left( 1 - \frac{1}{2n} \right) \right)$$

$$\leq \tan(\theta_0)^2 \|P\|_\infty^2 \left( 1 - \frac{1}{2n - \sqrt[2n]{2n}} \left( 1 - \frac{1}{2n} \right) \right)$$

It is easy to show that

$$|\hat{J}(K) - \|T_N(P)\|| \leq \tan(\theta_0)^2 \|P\|_\infty \left( 1 - \frac{1}{2n - \sqrt[2n]{2n}} \left( 1 - \frac{1}{2n} \right) \right).$$

In addition, for $n \geq 1$, one can verify that

$$1 - \frac{1}{2n - \sqrt[2n]{2n}} \left( 1 - \frac{1}{2n} \right) \leq n^{-\frac{2}{3}}$$

and hence $|\hat{J}(K) - \|T_N(P)\|| \leq C_3 \|P\|_\infty \frac{1}{\sqrt{n}} = \zeta_2(K)$ with $C_3 = \tan(\theta_0)^2$. This gives us sample complexity (16). Finally, applying triangle inequality, we have

$$|\hat{J}(K) - J(K)| \leq |\hat{J}(K) - \|T_N(P)\|| + |\|T_N(P) - J(K)\|| \leq \zeta_1(K) + \zeta_2(K) \tag{B.22}$$

which completes the proof.

### B.6  Proof of Theorem 4.2

We consider that we have an inaccurate function value estimation $\hat{J}(K) = J(K) + \zeta(K)$, here we assume that the estimation error $\zeta(K)$ is uniformly upper bounded for all $K \in \mathcal{S}^1$ as stated in (18). Then we have:

$$\begin{aligned}
\hat{g}^t &= \frac{d}{2\delta}(\hat{J}(K^t + \delta W^t) - \hat{J}(K^t - \delta W^t))W^t \\
&= \frac{d}{2\delta}(J(K^t + \delta W^t) - J(K^t - \delta W^t))W^t + \frac{d}{2\delta}(\zeta(K^t + \delta W^t) - \zeta(K^t - \delta W^t))W^t \\
&= g^t + \frac{d(\zeta_1^t - \zeta_2^t)}{2\delta}W^t,
\end{aligned}$$

where we denote $\zeta_1^t = \zeta(K^t + \delta W^t)$ and $\zeta_2^t = \zeta(K^t - \delta W^t)$ for simplicity. To prove Theorem 4.2, we first present some technical lemmas.

**Lemma B.10.** *Suppose that $J(K)$ is $L_1$-Lipschitz on the sublevel set $\mathcal{S}^1$ and let $\{g^t\}_{t=0}^{T-1}$ and $\{K^t\}_{t=0}^{T-1}$ be generated by Algorithm 1 such that $\{K^t\}_{t=0}^{T-1}$ are feasible and $\{J_\delta(K^t)\}_{t=0}^{T-1}$ are well defined. Then, we have*

$$\|\mathbb{E}[\hat{g}^t \mid \mathcal{F}_t] - \nabla_\delta J(K^t)\|_F \leq \frac{d\kappa}{\delta}, \tag{B.23}$$

$$\mathbb{E}[\|\hat{g}^t\|_F^2 \mid \mathcal{F}_t] \leq 32\sqrt{2\pi}dL_1^2 + 2(\frac{d\kappa}{\delta})^2. \tag{B.24}$$

*Proof.* By definition of $\hat{g}^t$, we have:

$$\|\mathbb{E}[\hat{g}^t \mid \mathcal{F}_t] - \nabla_\delta J(K^t)\|_F = \|\mathbb{E}\left[g^t + \frac{d(\zeta_1^t - \zeta_2^t)}{2\delta}W^t \mid \mathcal{F}_t\right] - \nabla_\delta J(K^t)\|_F$$

$$= \|\mathbb{E}\left[g^t \mid \mathcal{F}_t\right] + \mathbb{E}\left[\frac{d(\zeta_1^t - \zeta_2^t)}{2\delta}W^t \mid \mathcal{F}_t\right] - \nabla_\delta J(K^t)\|_F$$

$$= \|\mathbb{E}\left[\frac{d(\zeta_1^t - \zeta_2^t)}{2\delta}W^t \mid \mathcal{F}_t\right]\|_F$$

$$\leq \mathbb{E}\left[\|\frac{d(\zeta_1^t - \zeta_2^t)}{2\delta}\|_F\|W^t\|_F \mid \mathcal{F}_t\right] \leq \frac{d\kappa}{\delta}.$$

$$\mathbb{E}[\|\hat{g}^t\|_F^2 \mid \mathcal{F}_t] = \mathbb{E}\left[\|g^t + \frac{d(\zeta_1^t - \zeta_2^t)}{2\delta}W^t\|_F^2 \mid \mathcal{F}_t\right]$$

$$\leq 2\mathbb{E}[\|g^t\|_F^2 \mid \mathcal{F}_t] + 2\mathbb{E}[\|(\frac{d(\zeta_1^t - \zeta_2^t)}{2\delta})W^t\|_F^2 \mid \mathcal{F}_t]$$

$$\leq 32\sqrt{2\pi}dL_1^2 + 2(\frac{d\kappa}{\delta})^2.$$

This completes the proof. $\qquad\square$

**Lemma B.11.** *$J(K)$ is $L_1$-Lipschitz on the sublevel set $\mathcal{S}^1$, let $\eta \leq \frac{\delta\xi}{d\kappa(100J(K^0)-J^*)}$ and $\delta \leq \min\{\Delta_1, \Delta\}$, then as long as $K^t \in \mathcal{S}^0$, we will have $K^{t+1} \in \mathcal{S}^1$ and*

$$\mathbb{E}[J_\delta(K^{t+1}) \mid \mathcal{F}_t] \leq J_\delta(K^t) - \eta\|\nabla J_\delta(K^t)\|_F^2 + \eta(Z_1 + Z_2), \tag{B.25}$$

*where $Z_1 = \frac{L_1 d\kappa}{\delta}$ and $Z_2 = \frac{\eta}{\delta}\Gamma$ with $\Gamma = cL_1\sqrt{d} \cdot (16\sqrt{2\pi}dL_1^2 + (\frac{d\kappa}{\delta})^2)$.*

*Proof.* Since $\|W^t\|_F = 1$, we have:

$$\|K^{t+1} - K^t\|_F = \eta\|\frac{d}{2\delta}(\hat{J}(K^t + \delta W^t) - \hat{J}(K^t - \delta W^t))W^t\|_F$$

$$\leq \frac{\eta d}{2\delta}|(J(K^t + \delta W^t) - J(K^t - \delta W^t)| \cdot |\zeta_1^t - \zeta_2^t|$$

$$\leq \frac{\eta d\kappa}{\delta} \cdot (100J(K^0) - J^*)$$

$$\leq \xi,$$

where the second inequality holds since we have $K^t \pm \delta W^t \in \mathcal{S}^1$ when $\delta \leq \Delta$ and $|\zeta_1^t - \zeta_2^t| \leq 2\kappa$. This implies that $K^{t+1} \in \mathcal{S}^1$ by Remark 3.5. In addition, $K^{t+1} \in \mathcal{S}^1$ and $\delta \leq \Delta_1$ ensures that $J_\delta(K^{t+1})$ is well defined. By Proposition 2.3 (ii), we know that $J_\delta(K)$ is differentiable and $L_1$-Lipschitz with $\frac{cL_1\sqrt{d}}{\delta}$-Lipschitz gradient on the sublevel set $\mathcal{S}^1$. Then we have:

$$J_\delta(K^{t+1}) \leq J_\delta(K^t) - \eta\langle\nabla J_\delta(K^t), \hat{g}^t\rangle + \frac{c\eta^2 L_1\sqrt{d}}{2\delta}\|\hat{g}^t\|_F^2.$$

Taking the expectation of both sides conditioned on $K^t$, we have

$$\mathbb{E}[J_\delta(K^{t+1}) \mid \mathcal{F}_t] \leq J_\delta(K^t) - \eta\langle\nabla J_\delta(K^t), \mathbb{E}[\hat{g}^t \mid \mathcal{F}_t]\rangle + \frac{c\eta^2 L_1\sqrt{d}}{2\delta}\mathbb{E}[\|\hat{g}^t\|_F^2 \mid \mathcal{F}_t]$$

$$\leq_{(i)} J_\delta(K^t) - \eta\langle\nabla J_\delta(K^t), \nabla J_\delta(K^t) - \nabla J_\delta(K^t) + \mathbb{E}[\hat{g}^t \mid \mathcal{F}_t]\rangle + \frac{\eta^2}{\delta}\Gamma$$

$$\leq_{(ii)} J_\delta(K^t) - \eta\|\nabla J_\delta(K^t)\|_F^2 + \eta\|\nabla J_\delta(K^t)\|_F\|\nabla J_\delta(K^t) - \mathbb{E}[\hat{g}^t \mid \mathcal{F}_t]\|_F + \frac{\eta^2}{\delta}\Gamma$$

$$\leq_{(iii)} J_\delta(K^t) - \eta\|\nabla J_\delta(K^t)\|_F^2 + \eta\frac{L_1 d\kappa}{\delta} + \frac{\eta^2}{\delta}\Gamma$$

$$= J_\delta(K^t) - \eta\|\nabla J_\delta(K^t)\|_F^2 + \eta Z_1 + \eta Z_2,$$

where inequality (i) and (iii) use the bounds in Lemma B.10, and (ii) holds by Cauchy-Schwarz inequality. This completes the proof. $\square$

Now we are ready to prove the Theorem 4.2. We will first prove Statement 1 in Theorem 4.2: all the generated controllers are within the feasible set with high probability. Then we will show Statement 2: the Algorithm 1 returns a $(\delta, \epsilon)$-stationary point with high probability.

**Proof of Statement 1** We first define a stopping time $\tau$ as below:

$$\tau := \min\{t \in \{0, 1, 2, \cdots, T-1\} : J_\delta(K^t) > 49J(K^0)\}. \tag{B.26}$$

Based on Lemma 3.2, it can be seen that as long as $\tau \geq T - 1$, the iterates $\{K^t\}_{t=0}^{T-1}$ generated by Algorithm 1 are feasible. Therefore our goal becomes bounding the probability $\Pr(\tau \leq T - 1)$. To this end, we can define a nonnegative supermartingale $Y(t)$ as below

$$Y(t) := J_\delta(K^{\min\{t,\tau\}}) + \eta(T-t)(Z_1 + Z_2). \tag{B.27}$$

To show it is a supermartingale, noticing that we have

$$\mathbb{E}[Y(t+1) \mid \mathcal{F}_t] = \mathbb{E}[J_\delta(K^\tau)\mathbb{1}_{\tau\leq t} \mid \mathcal{F}_t] + \mathbb{E}[J_\delta(K^{t+1})\mathbb{1}_{\tau>t} \mid \mathcal{F}_t] + \eta(T-t-1)(Z_1 + Z_2)$$

$$= J_\delta(K^\tau)\mathbb{1}_{\tau\leq t} + \mathbb{E}[J_\delta(K^{t+1})\mathbb{1}_{\tau>t} \mid \mathcal{F}_t] + \eta(T-t-1)(Z_1 + Z_2)$$

$$\leq^{(i)} J_\delta(K^\tau)\mathbb{1}_{\tau\leq t} + J_\delta(K^t)\mathbb{1}_{\tau>t} - \eta\|\nabla J_\delta(K^t)\|_F^2 + \eta(Z_1 + Z_2) + \eta(T-t-1)(Z_1 + Z_2)$$

$$\leq J_\delta(K^{\min\{t,\tau\}}) + \eta(T-t)(Z_1 + Z_2) = Y(t),$$

where the inequality $(i)$ uses (B.25). This is valid since $\tau > t$ implies $J_\delta(K^t) \leq 49J(K^0)$ by the definition of $\tau$. Then we have $J(K^t) \leq J_\delta(K^t) + \delta L_1 \leq 50J(K^0)$. Hence we have $K^t \in \mathcal{S}^0$. The choice of $\eta$ guarantees that $K^{t+1} \in \mathcal{S}^1$. Hence $K^{t+1}$ is feasible and $J_\delta(K^{t+1})$ is well defined when $\delta \leq \Delta_1$. Then Doob's maximal inequality for super-martingales gives

$$\Pr(\tau \leq T-1) \leq \Pr(\max_{t=0,1,\cdots,T-1} Y(t) > 49J(K^0))$$

$$\leq \frac{\mathbb{E}[Y(0)]}{49J(K^0)} = \frac{J_\delta(K^0) + T\eta(Z_1 + Z_2)}{49J(K^0)} \leq \frac{J(K^0) + \delta L_1}{49J(K^0)} + \frac{T\eta(Z_1 + Z_2)}{49J(K^0)}$$

$$\leq \frac{2}{49} + \frac{T\eta(Z_1 + Z_2)}{49J(K^0)}$$

For sufficiently small $\epsilon$, we have $\eta = \frac{\delta\epsilon^2}{100\Gamma}$, then one can verify that $T\eta(Z_1 + Z_2) \leq \upsilon J(K^0)$ by the choice of $T, \eta,$ and $\kappa$. Therefore, we have

$$\Pr(\tau \leq T-1) \leq \frac{2}{49} + \frac{T\eta Z}{49J(K^0)} \leq \frac{2}{49} + \frac{\upsilon}{49} \leq 0.05 + 0.03\upsilon.$$

This implies that all the iterates $K^t$ are stabilizing with probability at least $1 - (0.05 + 0.03\upsilon) = 0.95 - 0.03\upsilon$. This completes the proof for Statement 1.

**Proof of Statement 2**   We first show that we can extend (B.25) as

$$\mathbb{E}[J_\delta(K^{t+1})\mathbb{1}_{\tau>t+1} \mid \mathcal{F}_t] \leq J_\delta(K^t)\mathbb{1}_{\tau>t} - \eta\|\nabla J_\delta(K^t)\|_F^2\mathbb{1}_{\tau>t} + \eta Z_1 + \eta Z_2, \qquad \text{(B.28)}$$

If $\tau > t$, then we know $J_\delta(K^t) \leq 49J(K^0)$, hence $J(K^t) \leq J_\delta(K^t) + \delta L_1 \leq 50J(K^0)$, we have $K^t \in \mathcal{S}^0$ and $K^{t+1} \in \mathcal{S}^1$. Therefore, we can apply Lemma B.3 to show that:

$$\mathbb{E}[J_\delta(K^{t+1})\mathbb{1}_{\tau>t+1} \mid \mathcal{F}_t] \leq \mathbb{E}[J_\delta(K^{t+1}) \mid \mathcal{F}_t]$$
$$\leq J_\delta(K^t) - \eta\|\nabla J_\delta(K^t)\|_F^2 + \eta Z_1 + \eta Z_2$$
$$= J_\delta(K^t)\mathbb{1}_{\tau>t} - \eta\|\nabla J_\delta(K^t)\|_F^2\mathbb{1}_{\tau>t} + \eta Z_1 + \eta Z_2.$$

On the other hand, if $\tau \leq t$, we have

$$\mathbb{E}[J_\delta(K^{t+1})\mathbb{1}_{\tau>t+1} \mid \mathcal{F}_t] = 0 \leq J_\delta(K^t)\mathbb{1}_{\tau>t} - \eta\|\nabla J_\delta(K^t)\|_F^2\mathbb{1}_{\tau>t} + \eta Z_1 + \eta Z_2 \qquad \text{(B.29)}$$

since $Z_1 \geq 0$ and $Z_2 \geq 0$. Therefore, (B.28) holds. Taking the expectation and rearranging the terms of (B.28) gives:

$$\mathbb{E}[\|\nabla J_\delta(K^t)\|_F^2\mathbb{1}_{\tau>t}] \leq \frac{\mathbb{E}[J_\delta(K^t)\mathbb{1}_{\tau>t}] - \mathbb{E}[J_\delta(K^{t+1})\mathbb{1}_{\tau>t+1}]}{\eta} + Z_1 + Z_2. \qquad \text{(B.30)}$$

Summing up the above inequality over $t = 0, 1, \cdots, T-1$ yields

$$\frac{1}{T}\sum_{t=0}^{T-1}\mathbb{E}[\|\nabla J_\delta(K^t)\|_F^2\mathbb{1}_{\tau>T-1}] \leq \frac{1}{T}\sum_{t=0}^{T-1}\mathbb{E}[\|\nabla J_\delta(K^t)\|_F^2\mathbb{1}_{\tau>t}]$$
$$\leq \frac{J_\delta(K^0) - \mathbb{E}[J_\delta(K^T)\mathbb{1}_{\tau>T}]}{\eta T} + Z_1 + Z_2.$$

Since the cost function $J(K)$ is nonnegative, we have $\mathbb{E}[J_\delta(K^T)\mathbb{1}_{\tau>T}] \geq 0$ and:

$$\frac{1}{T}\sum_{t=0}^{T-1}\mathbb{E}[\|\nabla J_\delta(K^t)\|_F^2\mathbb{1}_{\tau>T-1}] \leq \frac{J(K^0)}{\eta T} + Z_1 + Z_2.$$

By the choice of $\eta$ and $T$, the above inequality becomes:

$$\frac{1}{T}\sum_{t=0}^{T-1}\mathbb{E}[\|\nabla J_\delta(K^t)\|_F^2\mathbb{1}_{\tau>T-1}] \leq \frac{J(K^0)}{\eta T} + Z_1 + Z_2 \leq (\frac{2}{\upsilon}+1)Z_2 + Z_1 \leq (\frac{2}{\upsilon}+2)\frac{\epsilon^2}{100}.$$

Since the random count $R \in \{0, 1, 2, \cdots, T-1\}$ is uniformly sampled, we have

$$\mathbb{E}[\|\nabla J_\delta(K^R)\|_F^2\mathbb{1}_{\tau>T-1}] = \frac{1}{T}\sum_{t=0}^{T-1}\mathbb{E}[\|\nabla J_\delta(K^t)\|_F^2\mathbb{1}_{\tau>T-1}] \leq (\frac{2}{\upsilon}+2)\frac{\epsilon^2}{100},$$

which implies

$$\mathbb{E}[\|\nabla J_\delta(K^R)\|_F\mathbb{1}_{\tau>T-1}] \leq (0.15\upsilon^{-\frac{1}{2}} + 0.15)\epsilon.$$

Therefore, we have

$$\Pr(\|\nabla J_\delta(K^R)\|_F \geq \epsilon) = \Pr(\|\nabla J_\delta(K^R)\|_F \geq \epsilon, \tau > T-1) + \Pr(\|\nabla J_\delta(K^R)\|_F \geq \epsilon, \tau \leq T-1)$$
$$\leq \Pr(\|\nabla J_\delta(K^R)\|_F \geq \epsilon, \tau > T-1) + \Pr(\tau \leq T-1)$$
$$\leq \frac{1}{\epsilon}\mathbb{E}[\|\nabla J_\delta(K^R)\|_F\mathbb{1}_{\tau>T-1}] + \Pr(\tau \leq T-1)$$
$$\leq 0.15\upsilon^{-\frac{1}{2}} + 0.15 + 0.05 + 0.03\upsilon$$
$$= 0.15\upsilon^{-\frac{1}{2}} + 0.2 + 0.03\upsilon,$$

where the second inequality uses the Markov's inequality. By Proposition 2.3 (iii), we have $\nabla J_\delta(K^R) \in \partial_\delta J(K^R)$. This implies that

$$\Pr(\min\{\|H\|_F : H \in \partial_\delta J(K^R) \geq \epsilon\}) \leq \Pr(\|\nabla J_\delta(K^R)\|_F \geq \epsilon) \leq 0.15\upsilon^{-\frac{1}{2}} + 0.2 + 0.03\upsilon.$$

Therefore our algorithm can find a $(\delta, \epsilon)$-stationary point with probability at least $1 - (0.15\upsilon^{-\frac{1}{2}} + 0.2 + 0.03\upsilon) = 0.8 - 0.15\upsilon^{-\frac{1}{2}} - 0.03\upsilon$. This completes the proof for Statement 2.

## C Numerical Experiments

In this section, we provide more details on the numerical experiments. All the experiments are performed on a desktop computer with a 3.7 GHz Intel i5-9600K processor.

### C.1 Two-dimension example

In this subsection, we first present an example where the iterates of Algorthm 1 can be visualized in controller space directly. In particular, we consider the following system

$$A = \begin{bmatrix} 0.5 & 0 & -1 \\ -0.5 & 0.5 & 0 \\ 0 & 0 & 0.5 \end{bmatrix}, \quad B = \begin{bmatrix} 1 \\ 1 \\ 0 \end{bmatrix}, \quad C = \begin{bmatrix} 1 & 1 & 0 \\ 0 & 0 & 1 \end{bmatrix}. \tag{C.1}$$

The weight matrices $Q$ and $R$ are set as identity matrices with proper dimensions. In this case, we have $K = [k_1 \quad k_2] \in \mathbb{R}^{1\times2}$. Since $A$ is a stable matrix, we initialize $K^0 = [0 \quad 0]$. For Algorthm 1, we set $\eta = 1 \times 10^{-3}$, $\delta = 1 \times 10^{-4}$, and $\epsilon = 1 \times 10^{-3}$. The iterates and contour lines of the $\mathcal{H}_\infty$ norm are drawn in Figure 2. It can be seen that Algorithm 1 yields to a local minimum and after 5000 iterations, it converges to a controller $K^* = [-0.1429 \quad -0.6425]$.

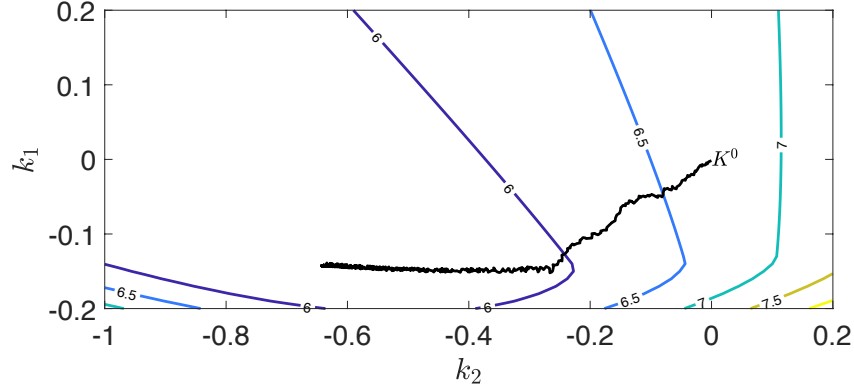

Figure 2: Algorithm 1 iterates in controller space for (C.1).

### C.2 Experiments for Figure 1

In this subsection, we discuss more details about the three different experiments displayed in Figure 1 to show the effectiveness of Algorithm 1 with exact and inexact oracle.

**Left plot of Figure 1**  For the left plot of Figure 1, we show the relative error of the Algorithm 1 with exact oracle for different system dimensions $n_x = \{10, 50, 100\}$. The system is of form (1) with parameters $(A, B, C)$. The entries of $(A, B, C)$ are sampled from a standard normal distribution $\mathcal{N}(0, 1)$. The system matrix $A$ is scaled when necessary to make $\rho(A) < 1$ such that we can set $K^0$ as zero matrices. The cost matrices $Q$ and $R$ are set to be identities with appropriate dimensions. Table 2 shows the detailed algorithm parameters.

Table 2: Algorithm 1 parameters

| $(n_x, n_u, n_y)$ | $\eta$ | $\delta$ | $\epsilon$ |
|---|---|---|---|
| $(10, 5, 5)$ | $1 \times 10^{-6}$ | $1 \times 10^{-4}$ | $1 \times 10^{-3}$ |
| $(50, 10, 10)$ | $1 \times 10^{-7}$ | $1 \times 10^{-4}$ | $1 \times 10^{-3}$ |
| $(100, 20, 20)$ | $1 \times 10^{-9}$ | $1 \times 10^{-4}$ | $1 \times 10^{-3}$ |

**Middle plot of Figure 1**  For the second experiment, we perform the Algorithm 1 with both exact and inexact zeroth-order oracles. In particular, we consider the following specific MIMO system:

$$A = \begin{bmatrix} 0.5 & 0 & -1 \\ -0.5 & 0.5 & 0 \\ 0 & 0 & 0.5 \end{bmatrix}, \ B = \begin{bmatrix} 1 & 0 \\ 0 & 1 \\ -1 & 0 \end{bmatrix}, \ C = \begin{bmatrix} 1 & 1 & 0 \\ 0 & 0 & 1 \end{bmatrix}. \tag{C.2}$$

The weight matrices $Q$ and $R$ are set as:

$$Q = \begin{bmatrix} 2 & -1 & 0 \\ -1 & 2 & -1 \\ 0 & -1 & 2 \end{bmatrix}, \ R = \begin{bmatrix} 1 & 0 \\ 0 & 1 \end{bmatrix}.$$

Since the system matrix $A$ is Schur stable, we set the initial point $K^0$ as a zero matrix. We also set the smooth radius $\delta = 0.001$ and the stepsize $\eta = 0.0001$. It can be seen that the inexact oracle case works well and tracks the exact oracle scenario.

**Right plot of Figure 1**  Finally, we examine how the sample complexity varies with different $\epsilon$ for the exact oracle setting with system matrices (C.2). It is worth mentioning that in Algorithm 1, we cannot really determine whether an iterate $K^t$ is a $(\delta, \epsilon)$-stationary point or not since we cannot calculate the exact $\nabla J_\delta(K^t)$. In practice, we use the magnitude of $g^t$ as a stationarity criterion. In particular, we terminate the algorithm when the norm of $g^t$ is less than $\epsilon$.

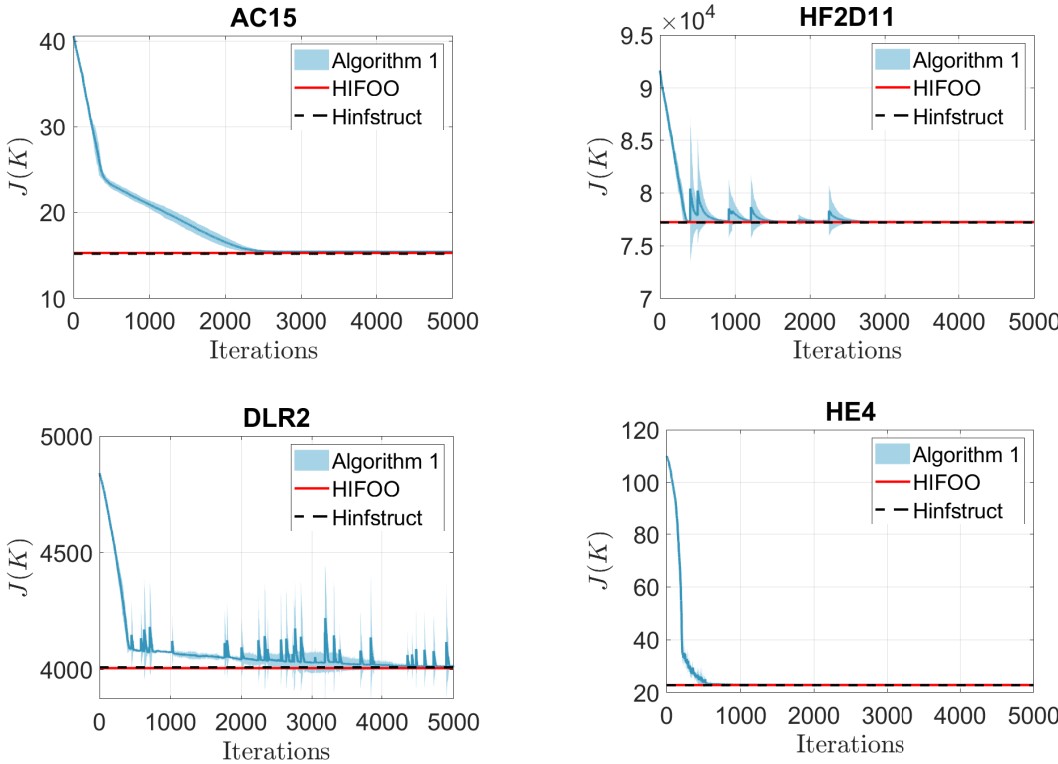

Figure 3: Algorithm 1 trajectories for selected the library examples.

## C.3  Comparison with model-based methods

In this subsection, we compare our derivative-free methods to model-based packages HIFOO and Hinfstruct, which are available in MATLAB. In particular, we select several benchmark models from COMPl$_e$ib including aircraft model AC15, 2D heat flow model HF2D11, second-order model DLR2, and helicopter models HE4 [41]. Table 3 summarized more details on the selected examples.

It is worth mentioning that the models in COMPl$_e$ib are all continuous time models. In addition, the implementations of HIFOO and Hinfstruct are also based on continuous-time models. Therefore, we

Table 3: Selected examples form COMPl$_e$ib

| Physical model | Example | $(n_x, n_u, n_y)$ | Structure of $A$ | Stability of $A$ |
|---|---|---|---|---|
| Air craft model | AC15 | $(4, 2, 3)$ | dense | stable |
| 2-D heat flow model | HF2D11 | $(5, 2, 3)$ | dense | unstable |
| second-order model | DLR2 | $(40, 2, 2)$ | sparse | stable |
| helicopter model | HE4 | $(8, 4, 6)$ | dense | unstable |

implemented the continuous-time version of Algorithm 1. In other words, when we compute the $\mathcal{H}_\infty$ norm of the closed-loop system, we are computing the $\mathcal{H}_\infty$ norm of a continuous-time system instead of a discrete-time system. Figure 3 shows the trajectories of the four selected examples when we run Algorithm 1 for 5000 iterations. It can be seen that our algorithm converges to the ones computed by the model-based methods HIFOO and Hinfstruct with proper initialization. In practice, one can use multiple initial points and run Algorithm 1 multiple times then just report the best case with the lowest $J(K)$. Such a strategy is used in HIFOO and Hinfstruct. It can be seen that our derivative-free method can achieve comparable results even without the knowledge of the system dynamics.

