# OpenReview forum: "Complexity of Derivative-Free Policy Optimization for Structured $\mathcal{H}_\infty$ Control"
_NeurIPS.cc/2023/Conference — NeurIPS 2023 poster_

### Official Review · Reviewer_UPcn · 2023-07-06

**Soundness:** 3 good
**Presentation:** 3 good
**Contribution:** 3 good
**Rating:** 6
**Confidence:** 3

**Summary:**

This paper considers solving the $H_\infty$ control problem using zero-th-order policy optimization. The main results are sample complexity bounds for both the exact Oracle setting and the model-free setting. Numerical simulations are conducted to demonstrate the effectiveness of the algorithm.

**Strengths:**

This paper is well-written. The $H_\infty$ control problem is known to be challenging and the policy optimization results are likely to be of great interest to the learning-in-control community. The sample complexity result, while not necessarily optimal, is the first non-asymptotic result in the literature.

**Weaknesses:**

(1) Related Work: The related work section could be more structured. I went through [32] to complete the review of this work. It seems that the main technical tools are already developed in [32]. However, [32] does not provide any sample complexity result. What are the major technical challenges (and the ideas used to overcome them) in going beyond the global convergence in [32] to the sample complexity results in this work?

(2) Theorem 3.7 and Theorem 4.2: Usually for high probability bounds, as the tolerance level $v$ decreases, more iterations are needed. However, for Theorem 3.7 and Theorem 4.2, as $v$ decreases, $T$ also decreases, which seems counter-intuitive. Moreover, the bound has a polynomial tail rather than an exponential tail. Is this an artifact of the proof or are exponential tail bounds not achievable?

**Questions:**

From the numerical simulations, is it possible to tell what the convergence rate of the algorithm is?

**Limitations:**

The sample complexity bounds are not necessarily optimal, which was pointed out by the authors.

---

> ### Author Rebuttal · Authors · 2023-08-10
>
> We thank the reviewer for taking time and effort to review our manuscript. We sincerely appreciate all your valuable comments and suggestions. Please see our responses below.
>
> **Related Work: The related work section could be more structured. What are the major technical challenges (and the ideas used to overcome them) in going beyond the global convergence in [32]?**
>
> We thank the reviewer for the valuable comment. For the $H_\infty$ control problem discussed in [32], the policy update is performed using the minimum-norm element of the associated Goldstein’s subdifferential. The main idea is based on the fact that such a minimum-norm element is a descent direction of the original cost function $J(K)$. While in our work, we aim to obtain the sample complexity of the zeroth-order methods for solving structured $H_\infty$ control problem. To this end, the zeroth-order oracle is used to construct an estimator of $\nabla J_\delta (K)$, which is a descent direction of the smoothed function $J_\delta (K)$. We obtain our sample complexity results based on the fact that an $\epsilon$-stationary point of $J_\delta (K)$ is also a $(\delta, \epsilon)$-stationary point of the original nonsmooth cost function $J(K)$. Indeed, there are several technical challenges in developing the sample complexity via this approach:
>
> 1. **Feasibility of the generated controllers.** In our control setup, unlike the unconstrained optimization problems, we need to ensure that the iterate $K^t$ and the perturbed iterate $K^t \pm \delta W^t$ stay within a non-convex feasible set (namely, the set of stabilizing policies). Previous work on policy optimization theory of $H_\infty$ control addresses this feasibility issue via using the coerciveness of $J(K)$ and mainly relies on the fact that $J(K)$ is a barrier function on the non-convex set of stabilizing policies. Such previous results rely on model-based algorithms (such as Goldstein's subgradient method) which can decrease the value of $J(K)$ directly. In our paper, we consider the model-free setting and hence need to use zeroth-order policy optimization. However, the zeroth-order policy search can only decrease the value of the smoothed function $J_\delta(K)$, which is not coercive over the non-convex feasible set and hence cannot be used as a barrier function. Importantly, the descent of $J_\delta(K)$ does not imply the descent of the original function value and hence does not ensure feasibility by itself. As a matter of fact, how to choose $\delta$ to ensure that $J_\delta$ is well defined is already non-trivial. Consequently, we need to design the smooth radius $\delta$ carefully (as indicated in Theorems 3.6, 3.7, and 4.2) to ensure the feasibility of the iterates.
>
> 2. **Inexact Oracle.** For Theorem 4.2, we consider the inexact oracle case which is particularly relevant for the model-free control setting. Specifically, we are using imperfect estimates of $J(K)$ that are calculated using the model-free MIMO power iteration method. Therefore, an extra statistical error term appears in the iterations of zeroth-order policy optimization, and requires special treatment. Such an extra term has not been considered in the literature of zeroth-order optimization for nonconvex nonsmooth problems. To address this extra technical difficulty, we first establish sample complexity bounds for $H_\infty$ norm estimation of the general MIMO system. Then we carefully propagate such sample complexity bounds to obtain an error bound for $\nabla J_\delta$ in terms of $\epsilon$ and $\delta$.
>
> We will include a more structured related work section and highlight our contributions in the revised manuscript. We emphasize that our study is more relevant to the model-free learning-based control setting, since our sample complexity results address the model-free control directly.
>
> **Theorem 3.7 and Theorem 4.2: Probability bounds seem counter-intuitive. Moreover, the bound has a polynomial tail rather than an exponential tail. Is this an artifact of the proof or are exponential tail bounds not achievable?**
>
> We thank the reviewer for such an insightful comment. We agree with the reviewer that in general, more iterations are needed to decrease the tolerance level. Our results are also intuitive. Specifically, in both Theorem 3.7 and 4.2, Statement 1 suggests that as $T$ increases, the probability that all the generated controllers are stabilizing will decrease. This is because our algorithm uses zeroth-order oracle to build an estimator of the smoothed function gradient. As $T$ increases, the biases and variance of the gradient estimation accumulate, resulting in a larger failure probability. In addition, Statement 2 suggests that as $T$ increases, the probability of finding a $(\delta, \epsilon) $-stationary point will first increase and then decrease. This is also intuitive, when $T$ is too small, more iterations will improve the performance of the generated controllers, but for large $T$, the probability of generating unstable controllers becomes dominant. We will add the above discussions in the revised manuscript.
>
> The reviewer is right about the polynomial tail about our bounds. This is the result of the proof techniques, and our polynomial bounds match the existing one for unconstrained optimization.
>
>  **Convergence rate from the numerical simulations.**
>
> From the left plot of Figure 1,  the convergence rate in terms of iteration vs. error looks sublinear.  This is consistent with our complexity theory which can be viewed as a "sublinear rate" result.

---

> > ### Comment · Reviewer_UPcn · 2023-08-14
> > **Acknowledgement of the Rebuttal**
> >
> > Thank the authors for their detailed response. I do not have further questions.

---

### Official Review · Reviewer_vRAL · 2023-07-06

**Soundness:** 3 good
**Presentation:** 2 fair
**Contribution:** 3 good
**Rating:** 6
**Confidence:** 3

**Summary:**

This paper focuses on the structured $H_\infty$ control problem. They provide sample complexity bounds for policy optimization in $H_\infty$ control problem.
The results are provided for two separate scenarios namely:
- Exact Oracle Setting  (exact $J(K)$ for any $K$ is available, for the given closed loop system)
- Inexact Oracle Setting (the system matrices are not known)

The theoritical results provide the sample complexity of $H_\infty$ norm estimation.
Finally, the paper provides a few numerical experiments supporting their theoritical results along with comparison to some model-based approaches.

**Strengths:**

- $H_\infty$ control problem is one of the important setting in linear systems , which is well studied in adaptive control literature but has received less attention in recent learning theory literature unlike the standard LQR setting. The paper highlights the various challenges involved in the analysis of $H_\infty$ control due to non-convexity and non-smoothness.

- The Sample complexity of the $H_\infty$ norm estimation are provided by exploiting the randomized smoothing techniques.



**Weaknesses:**

- The algorithm relies on access to an oracle. More discussion on the oracle is warranted.
- The simulation section in the main body (as well as the appendix) of the paper is meager. The presentation quality of plots included can be improved.
- Authors mention in the appendix that (line 806) that when necessary, one can reinitiate to avoid bad local minima, but how would one know that they are at a 'bad' local minima.


**Questions:**

Following are my concerns/ questions regarding the paper.
- Can authors explain why they call this approach as "derivative free" approach since calculating $g_t$  is required in the algorithm?
- I would like to know the motivation behind considering the two separate type of oracle scenarios.
- Authors mention that their probability bounds can be sharpened (line 248), I request authors to present the best possible results in the main body of the paper.

I also request authors to address the comments in the weakness section if possible.


**Limitations:**

The results rely on access to an oracle.

Social Impact: NA

---

> ### Author Rebuttal · Authors · 2023-08-10
>
> We thank the reviewer for taking time and effort to review our manuscript. We sincerely appreciate all your valuable comments. Please see our responses below.
>
> **More discussion on the oracle is warranted.**
>
> We appreciate the valuable suggestion from the reviewer. Our paper considers two zeroth-order oracles: the exact oracle is standard for zeroth-order optimization literature and natural for the model-based control setting, while the inexact oracle is relevant for the model-free learning-based control setting. The exact oracle assumes that we can exactly calculate $J(K)$ (which is the closed-loop $H_\infty$ norm) for every stabilizing $K$.  When the system dynamics are known, such an oracle is available since the closed-loop $H_\infty$ norm can be efficiently calculated using existing robust control packages in MATLAB (currently, the state-of-the-art techniques for model-based $H_\infty$ norm calculations rely on using the relation between the singular values of the transfer function matrix and the eigenvalues of a related Hamiltonian matrix [BBK1989,BS1990]).
>
> [BBK1989] Boyd, Balakrishnan, and Kabamba, 1989. A bisection method for computing the $H_\infty$ norm of a transfer matrix and related problems. Mathematics of Control, Signals and Systems.
> [BS1990] Bruinsma and Steinbuch, 1990. A fast algorithm to compute the $H_\infty$-norm of a transfer function matrix. Systems & Control Letters.
>
> However, in the model-free learning-based $H_\infty$ control setting, the system dynamics are unknown, and $J(K)$ (the closed-loop $H_\infty$ norm) can only be estimated from the input/output data of a black-box simulator of the underlying system. The inexact oracle is natural for such a model-free setting and can be provided by model-free $H_\infty$ norm estimations methods (see the beginning of Section 4.1 for a review). Our paper uses the model-free time-reversal power-iteration-based $H_\infty$ estimation from [WSH2010] to serve as the inexact oracle for $J(K)$. Despite the existence of such algorithms, the prior literature lacks sample complexity bounds for general MIMO systems. Therefore, we first present the first sample complexity result for $H_\infty$ norm estimation for general MIMO systems in Theorem 4.1. Building upon this, we obtain the first sample complexity results for model-free policy optimization of $H_\infty$ control with noisy function values. We will add more discussion on this in the revised manuscript.
>
> [WSH2010] Wahlberg, Syberg, and Hjalmarsson, 2010. Non-parametric methods for $\ell_2$-gain estimation using iterative experiments. Automatica.
>
> **The simulation section in the main body (as well as the appendix) of the paper is meager. The presentation quality of plots included can be improved.**
>
> We thank the reviewer for the constructive comments. We agree with the reviewer that the presentation quality of the plots should be improved. We have revised the plots via proper scaling and adding more explanations. Please see our uploaded one-page pdf file. If the reviewer has more concrete suggestions, please let us know. Any comments are highly appreciated and we will revise accordingly.
>
> **How would one know that they are at a 'bad' local minimum.**
>
> That statement is misleading and we will revise it. In practice, one can use multiple initial points and run the algorithm multiple times then just report the best case with the lowest $J(K)$. Such a strategy is used in existing packages such as HIFOO and Hinfstruct. So one does not explicitly check whether the solution is a bad local minimum or not. In general, there are no polynomial-time guarantees for finding the global solutions of such nonconvex problems.
>
> **Can authors explain why they call this approach as "derivative free" approach since calculating $g_t$ is required in the algorithm?**
>
> In Algorithm 1, we use the function value $J(K)$ to build an estimation of $\nabla J_\delta J(K)$. The function value is typically referred to as “zeroth-order oracle” in zeroth-order optimization literature. The methods that use zeroth-order oracle are referred to as derivative-free methods or zeroth-order methods [GL2013, CSV2009].
>
> [GL2013] Ghadimi and Lan, 2013. Stochastic first-and zeroth-order methods for nonconvex stochastic programming. SIAM Journal on Optimization.
> [CSV2009] Conn, Scheinberg, and Vicente, 2009. Introduction to derivative-free optimization. SIAM
>
> **Motivation behind considering the two separate types of oracle scenarios.**
>
> We first study the exact oracle case, which is a standard oracle assumption in zeroth-order optimization literature. This oracle can be obtained when the system matrices are known (model-based oracle). Building upon the complexity results under this assumption, we further extend our analysis to a more practical context: the inexact oracle scenario, where we only have access to a black-box system simulator for generating imperfect estimates of $J(K)$.  This aligns more closely with learning-based control where system models are unknown (model-free oracle).
>
> **The best possible results in the main body of the paper.**
>
> We are sorry for the confusion here. We really mean that the constant factors of the probability bounds in Theorems 3.7 and 4.2 can be improved by e.g., increasing the level of $\mathcal{S}^1$, using smaller step sizes, using smaller smooth radius delta, etc. For example, in the proof of Theorem 3.7, if we choose a larger sub level of the set $\mathcal{S}^1$, then we can obtain a refined constant term in the probability bounds.  However, we want to emphasize that refining the constant factors will not change the order dependence of $\epsilon$ and $\delta$ in the sample complexity results (Eq. 12, Eq. 21), which stands as the most crucial aspect of the sample complexity theory. The order of our curret sample complexity results is already the best as we can find. We will clarity this point in revision.

---

> > ### Comment · Reviewer_vRAL · 2023-08-10
> > **Response to Author Rebuttal**
> >
> > I thank the authors for their responses and incorporating the modification regarding the presentation of simulations. After reading their rebuttal, I will retain my recommendation regarding this paper.

---

### Official Review · Reviewer_v5nm · 2023-07-27

**Soundness:** 3 good
**Presentation:** 3 good
**Contribution:** 2 fair
**Rating:** 5
**Confidence:** 2

**Summary:**

This paper studies the static output feedback $\mathcal{H}_{\infty}$ control problem. It proposes a derivative-free policy optimization algorithm via randomized smoothing and further provides sample complexity analysis for the cases with exact and inexact zeroth-order oracles. To validate the performance of the new algorithm, the authors also conduct some numerical experiments and compete against the model-based methods in the literature.

**Strengths:**

1. As the authors claim, the proposed algorithm is the first derivative-free policy optimization algorithm for constrained structured $\mathcal{H}_{\infty}$ control problem (there have been some works on the unconstrained setting).
2. The authors also consider the inexact oracle setting.
3. The paper provides both theoretical analysis and numerical experiments.

**Weaknesses:**

I am not very familiar with the problem studied by this paper and it looks fine to me. However, I feel like the paper is mainly a combination of existing techniques (such as randomized smoothing and gradient sampling with zeroth order feedback). Would the authors highlight any novel techniques they apply in the analysis or explain what makes the problem different from other nonsmooth nonconvex problems such that this paper is not simply A+B?

**Questions:**

None.

**Limitations:**

None.

---

> ### Author Rebuttal · Authors · 2023-08-08
>
> We appreciate the reviewer's time and effort in evaluating our manuscript. We value your insightful comments and suggestions. Below are our responses.
>
> **Would the authors highlight any novel techniques they apply in the analysis or explain what makes the problem different from other nonsmooth nonconvex problems such that this paper is not simply A+B?**
>
> We thank the reviewer for the valuable comment. We would like to highlight the novelty of our main results presented in Theorem 3.6, 3.7, and 4.2 in twofold:
>
> 1. **Feasibility of the generated controllers.** In our control setup, unlike the unconstrained optimization problems, we need to ensure that the iterate $K^t$ and the perturbed iterate $K^t \pm \delta W^t$ stay within a non-convex feasible set (namely, the set of stabilizing policies). Previous work on policy optimization theory of $H_\infty$ control addresses this feasibility issue via using the coerciveness of $J(K)$ and mainly relies on the fact that $J(K)$ is a barrier function on the non-convex set of stabilizing policies. Such previous results rely on model-based algorithms (such as Goldstein's subgradient method) which can decrease the value of $J(K)$ directly. In our paper, we consider the model-free setting and hence need to use zeroth-order policy optimization. However, the zeroth-order policy search can only decrease the value of the smoothed function $J_\delta(K)$, which is not coercive over the non-convex feasible set and hence cannot be used as a barrier function. Importantly, the descent of $J_\delta(K)$ does not imply the descent of the original function value and hence does not ensure feasibility by itself. As a matter of fact, how to choose $\delta$ to ensure that $J_\delta$ is well defined is already non-trivial. Consequently, we need to design the smooth radius $\delta$ carefully (as indicated in Theorems 3.6, 3.7, and 4.2) to ensure the feasibility of the iterates.
>
>
>
> 2. **Inexact Oracle.** For Theorem 4.2, we consider the inexact oracle case which is particularly relevant for the model-free control setting. Specifically, we are using imperfect estimates of $J(K)$ that are calculated using the model-free MIMO power iteration method. Therefore, an extra statistical error term appears in the iterations of zeroth-order policy optimization, and requires special treatment. Such an extra term has not been considered in the literature of zeroth-order optimization for nonconvex nonsmooth problems. To address this extra technical difficulty, we first establish sample complexity bounds for $H_\infty$ norm estimation of the general MIMO system. Then we carefully propagate such sample complexity bounds to obtain an error bound for $\nabla J_\delta$ in terms of $\epsilon$ and $\delta$.
> Built upon this, we demonstrate that Algorithm 1 remains effective even with an inexact oracle, ensuring the feasibility of the iterates while achieving finite-time sample complexity with high probability. The inexact zeroth-order oracle has not been considered by any previous papers.

---

> > ### Comment · Reviewer_v5nm · 2023-08-17
> >
> > Thanks for the response! I do not have further questions.

---

### Author Rebuttal · Authors · 2023-08-10

We deeply appreciate the insightful feedback provided by the reviewers. In response to the comments from Reviewer vRAL, we have attached a PDF file containing the updated plots from the main paper. Each comment from the reviewers has been addressed below. We hope our explanations have resolved the reviewers' concerns. Please let us know if you have any additional questions or require further clarification, all comments are highly appreciated.

---

### Decision · Program_Chairs · 2023-09-21

**Decision:**

Accept (poster)

**Comment:**

I appreciate the authors for conducting the rebuttal and the following discussions. Unlike most linear control sample complexity results that heavily rely on the fact that the cost functions in these benchmark problems are differentiable over the entire feasible set, this paper tackles the non-smooth $H_\infty$ control which can only access the zeroth-order oracle. The major contributions, the design for the smooth radius $J_\delta(K)$, iterations to ensure $K^t$ and $K^t+\delta W^t$ are stabilizing controllers, and the first sample complexity result via the model-free MIMO power iteration method, are significant to both NeurIPS and control communities. Therefore, I recommend the paper be accepted. Besides, as a small issue mentioned by reviewer UPcn, the authors should clearly discuss how the polynomial tail in the theorem compares to other control works and whether it is possible to further break it and attain the exponential tails in the final revision